# DECOMPOSED LEARNING AND GROKKING

## ABSTRACT

Grokking is a delayed transition from memorisation to generalisation in neural networks. It poses challenges for efficient learning, particularly in structured tasks and small-data regimes. This paper explores grokking in modular arithmetic, explicitly focusing on modular division with a modulus of 97. We introduce a novel learning method called Decomposed Learning, which leverages Singular Value Decomposition (SVD) to modify the weight matrices of neural networks. Decomposed learning reduces or avoids grokking by changing the representation of the weight matrix, $A$, into the product of three matrices $U$, $\Sigma$ and $V^T$, promoting the discovery of compact, generalisable representations early in the learning process. Through empirical evaluations on the modular division task, we show that Decomposed Learning significantly reduces the effect of grokking and, in some cases, eliminates it. Moreover, Decomposed Learning can reduce the parameters required for practical training, enhancing model efficiency and generalisation. These results suggest that our SVD-based method provides a practical and scalable solution for mitigating grokking, with implications for broader transformer-based learning tasks.

## 1 INTRODUCTION

Understanding the learning dynamics of over-parameterised deep learning models is a critical area of research, especially as these models are increasingly deployed in the real world. Although significant advances have been made in optimising training and improving performance, the finer details of how neural networks generalise remain a significant challenge with problems such as the capacity to memorise randomly labelled data (Zhang et al., 2017), deep double-decent (Nakkiran et al., 2019) and grokking (Power et al., 2022). The grokking phenomenon introduced by Power et al. (2022) is where models experience a significant delay between generalisation and the point at which a model achieves perfect training accuracy. This phenomenon suggests inefficiencies in how neural networks learn and training setups, which supports Liu et al. (2023), who show that the grokking phenomena can be induced on MNIST LeCun et al. (2010), IMDB dataset for sentiment analysis Maas et al. (2011). Although grokking can easily avoided for MNIST and IBMD datasets with correct training hyperparameters.

In this paper, we investigate the original grokking problem using the division mod 97 task, using a two layer transformer, aiming to understand better the underlying mechanisms that give rise to this delay in generalisation. We propose a novel approach that leverages Singular Value Decomposition (SVD) to change the representation of the weight matrices of the layers within the transformer model, aiming to mitigate and avoid delayed generalisation. By applying SVD, we decompose the weight matrix, $A$, into three matrices, $U_k$, $\Sigma_k$ and $V_k^T$, where $k$ is the rank, and explore how this representation, with different layers, ranks, and amounts of training data, affects the learning process, specifically delayed generalisation.

In this paper, we ask:

- How does the decomposed representation of the weight matrix, $A$, affect training?
- What is the relationship between the rank of a weight matrix and the amount of training data?
- How are different layers affected by the decomposition and rank?

The questions are explored in-depth, and our contributions are:

- Representing the weight matrix $A$ as the product of the three matrices $U_k$, $\Sigma_k$ and $V_k^T$ improves performance and can achieve superior results with fewer parameters in this grokking setup.

- As more training data is represented, fewer ranks are needed to mitigate or prevent the grokking phenomenon.

- Different layers can learn with varying degrees of rank reduction while preserving performance and reducing/avoiding grokking when using our SVD-based decomposed learning method.

## 2 RELATED WORK

### 2.1 GROKKING

Grokking is the name given to a phenomenon discovered by Power et al. (2022) when training a transformer architecture on binary operations beyond the point where full training accuracy is reached. Effectively, the training data is memorised (training accuracy is perfect or near-perfect), while test accuracy is at the random accuracy baseline; after significantly more training, the training performance is reached on the test set. Power et al. (2022) showed that increasing the training data can reduce the steps for grokking to occur. Liu et al. (2022) identified four learning phases that can occur during training, with grokking being one of them and could be avoided with specific hyperparameter tuning. Liu et al. (2023) showed that grokking can also occur on more complicated datasets such as image classification on MNIST. They attributed grokking to a discrepancy between training and test losses at a high model weight norm, referred to as the " LU" mechanism. Kumar et al. (2024) suggests that grokking can occur as the neural network transitions from lazy (linear) to rich (feature) learning. Miller et al. (2024) highlighted that grokking is not limited to neural networks and showed that it can occur with Gaussian processes, linear regression and Bayesian neural networks, suggesting that grokking may happen in any model where the solution is guided by complexity and error.

### 2.2 MATRIX DECOMPOSITION AND DEEP LEARNING

Despite neural networks typically being trained with all their parameters, they have been shown to have an intrinsic dimensionality (Li et al., 2018), allowing fewer parameters to be used to reach the same performance. Aghajanyan et al. (2020) showed that pre-trained language models have a low intrinsic dimensionality. Hu et al. (2022) introduced Low-Rank Domain Adaptation (LoRA) as a method for finetuning the self-attention module of large language models. This method trains a weight matrix as a rank-reduced composition of two matrices. Training on this decomposition allowed finetuning to use significantly fewer training parameters while achieving the same or better performance than conventional finetuning. Since methods based on similar approaches have been introduced, such as LoHa (Hyeon-Woo et al., 2023), LoKa (Edalati et al., 2022), OFT (Qiu et al., 2023). In addition to fine-tuning, matrix decompositions, specifically SVD, have been used to make training more efficient by performing low-rank projections on the gradient updates (Zhao et al., 2024; Zhang et al., 2024), compressing models with little performance degradation (Swaminathan et al., 2020; Liebenwein et al., 2021). SVD has also been used dynamically through training by Paul & Nelson (2021), who proposed a learning method using SVD on dense linear layers to reduce the rank progressively and, by extension, the dimensionality of the network during training. It introduces two hyperparameters, the threshold to remove singular values, and how often to perform this operation. This method reduced the training times up to 50% with minimal impact of accuracy on audio classification problems.

## 3 DECOMPOSED LEARNING

Neural network layers are represented by a weight matrix, $A$, that can represent a linear transformation such as rotation, scaling, shearing, reflection, or a combination. In this paper, instead of learning $A$ for each layer directly, we propose learning it in its decomposed or factorised form. To do this, we use the Reduced Singular Value Decomposition (SVD) outlined below.

SVD algorithmically decomposes a $m \times n$ matrix $A$ as the product of 3 matrices, $U$, $\Sigma$ and $V^T$ i.e.

$$A = U\Sigma V^T \tag{1}$$

where $U$ is $m \times r$, $\Sigma$ is $r \times r$ and $V^T$ is $r \times n$.

Each of these has specific properties: $U$ is comprised of $r$ orthonormal columns that span the columns space of $A$, $V$ is comprised of $r$ orthonormal columns that span the row space of $A$ and $\Sigma$ is diagonal - its entries are known as Singular Values, $\sigma_i$ (Strang, 2000).

We can further re-write A as a sum of simple matrices (known as rank one matrices) as

$$A = \sum_{i=1}^{r} u_i \sigma_i v_i^T \tag{2}$$

where r is the true rank of matrix $A (r \leq min(m, n))$ and $u_i$, $v_i$ are the columns of $U$ and $V$, respectively. The importance of SVD lies in the fact that it provides a way to approximate $A$, such that the computation of the layer activations can be reduced with a minimal loss in accuracy and representational power. This is known as reducing the rank of $A$. This is achieved by limiting the summation in (2) to $k < r$. Thus, the low-rank approximation is

$$\tilde{A}_k = U_k \Sigma_k V_k^T \tag{3}$$

where $U_k$ is $m \times k$, $\Sigma_k$ is an $k \times k$ diagonal matrix containing only the top $k$ largest singular values and $V_k^T$ is $k \times n$. $k$ is known as the rank.

In this paper, we propose and explore the effects of decomposing $A$ using SVD and training the matrix product in Eq (1) as well as using the decomposition in Eq (3) for various values of $k$. Thus, Decomposed Learning initialises the weight matrix, $A$, using standard methods, such as Xavier Uniform and Xavier Normal (Glorot & Bengio, 2010). The weight matrix, $A$, is then decomposed using SVD, providing initial values for $U$, $\Sigma$ and $V^T$ and the rank is reduced to create $U_k$, $\Sigma_k$ and $V_k^T$. Training proceeds without retaining the above-mentioned SVD properties of orthonormality and diagonality. Hence, for example, the trained $\Sigma_k$ can be non-diagonal. After training, $A^o$ is recomposed as

$$A^o = U_k^o \Sigma_k^o V_k^{To} \tag{4}$$

where $U_k^o$, $\Sigma_k^o$ and $V_k^{To}$ are the trained matrices of $U_k$, $\Sigma_k$ and $V_k^T$. See Figure 1 for a diagrammatic depiction of the process.

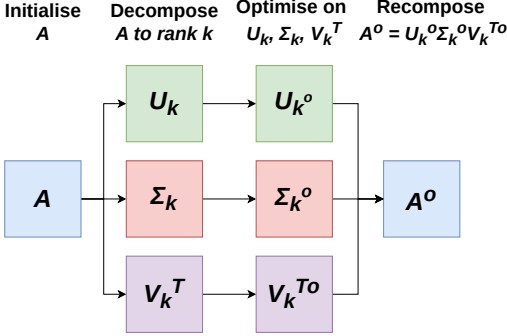

Figure 1: Decomposed Learning Process: Initialise $A$, decompose, $A$ to rank , $k$, with SVD into $U_k$, $\Sigma_k$ and $V_k^T$, train on $U_k$, $\Sigma_k$ and $V_k^T$ resulting in $U_k^o$, $\Sigma_k^o$ and $V_k^{To}$, recompose $A^o$ through a linear combination of the matrices $A^o = U_k^o \Sigma_k^o V_k^{To}$.

Training with a full rank-decomposition of $A$, i.e. $U$, $\Sigma$, and $V^T$, will increase the number of parameters of the layer compared to training directly on $A$. However, low-rank decompositions of $A$, with $U_k$, $\Sigma_k$ and $V_k^T$, can result in fewer training parameters depending on the rank, $k$. The equation equation 5, calculates the rank, $k$, that will result in approximately the same number of training parameters as training on $A$. Using a rank, $k$, lower than this value will result in fewer training parameters.

$$k = \left\lceil \frac{-(m+n) + \sqrt{(m+n)^2 - 4(mn)}}{2} > 0 \right\rceil \quad (5)$$

It is important to note that when either version is recomposed, the new matrix, $A^o$, will have the **same number of parameters** as the original weight matrix, $A$.

## 4 EXPERIMENTAL SETUP

Decomposed learning is explored in grokking using the division mod 97 task matching the original experimental setup by Power et al. (2022). This task is explored as it is the foundational grokking experiment. In addition, it is a complete algorithmic dataset that fully represents the problem space, meaning that training on $x\%$ of the dataset represents $x\%$ of the problem space. This property allows for a more precise investigation of how the amount of training data and rank affects the learning process. A 2-layer transformer decoder architecture (Vaswani et al., 2017) with a width of 128 and 4 attention heads is used for all the experiments as done by Power et al. (2022).

The effect of decomposed learning is explored with the following layers and ranks:

- Token Embedding Ranks: 12, 25, 50, 74 and **99**
- Position Embedding Ranks: 1, 2, 3, 4 and **5**
- Multi-Head Attention Ranks: 16, 32, 64, 96 and **128**
- Feed-forward Block Ranks: 16, 32, 64, 96 and **128**
- Output Layer Ranks: 12, 25, 50, 74 and **99**

The ranks represent approximately 12.5%, 25%, 50%, 75%, and 100% of the total ranks available for the respective layers. This selection allows for a broad understanding of how the rank can affect the model's ability to learn. The value in **bold** is the full rank representation. The feed-forward block rank decomposition is applied to both linear layers within the feed-forward block and both transformer blocks. See Figure 2 for a diagrammatic depiction of the explored layers.

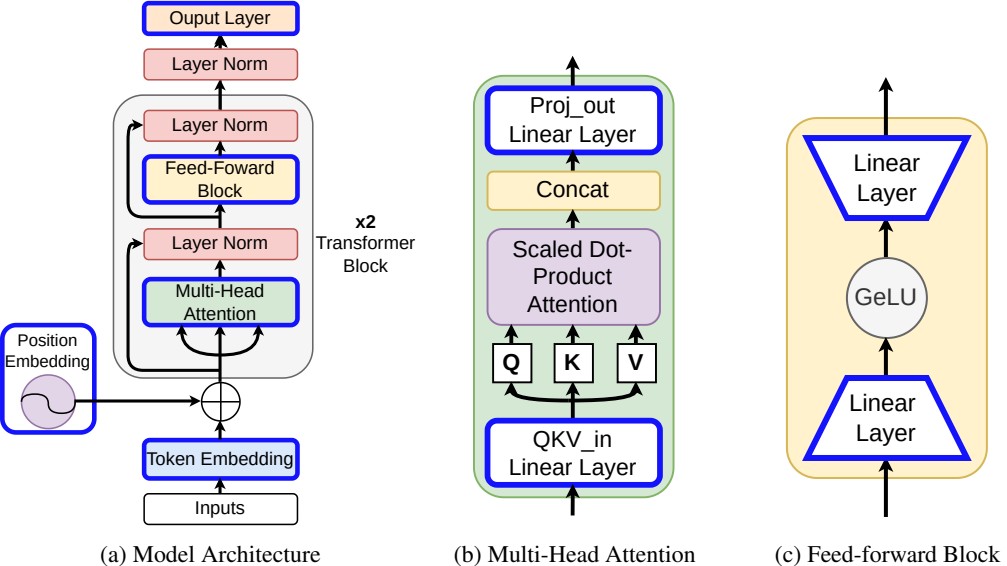

(a) Model Architecture     (b) Multi-Head Attention     (c) Feed-forward Block

Figure 2: Model architecture: sections with a dark blue border are investigated with decomposed learning. Figures adapted from (Vaswani et al., 2017)

The effect of dataset size on rank decomposition learning is explored using 50%, 65% and 80% of the dataset for training with $10^6$, $10^5$ and $10^4$ optimisation steps, respectively. For comparison,

a baseline model is trained typically with no layers decomposed. In all experiments, the model is trained using the AdamW optimiser (Loshchilov & Hutter, 2019) with a learning rate of 0.001, weight decay of 0, $\beta_1 = 0.9$ and $\beta_2 = 0.98$, a linear learning rate warm-up for the first ten iterations and a batch size of 512 as done in the original paper by Power et al. (2022).

Each layer is decomposed independently for all experiments, with all other layers trained normally unless otherwise stated; for example, in Section 5.1, the Token Embedding is decomposed, and all other layers are represented normally.

The mean of 5 runs is reported for all plots.

Additional experiential analysis of decomposed learning on grokking-induced MNIST (Liu et al., 2023) and real-world tasks on Tiny Shakespeare (Karpathy, 2015) and CIFAR 10 (Krizhevsky, 2009) datasets is provided in Appendix F and G respectively.

## 5 RESULTS

The results section compares normal training against decomposed learning on only the token embedding, 5.1, position embedding 5.2, multi-head attention 5.3, feed-forward blocks 5.4, output layer 5.5 and when decomposed learning on the token embedding, multi-head attention, feed-forward block and output layer altogether 5.6.

### 5.1 TOKEN EMBEDDING

The impact of decomposed learning on the token embedding layer at rank 12, 25, 50, 74 and full rank 99, compared to the normally trained model (baseline), is shown in Figure 3. This Figure, specifically Figure 3a, clearly shows the grokking phenomena with the training (dotted lines) increasing to a near-perfect accuracy between steps $10^2$ and $10^3$, while the test accuracy (solid lines) is still at random accuracy. Then significantly later, between steps $10^4$ and approximately $5 \times 10^5$, the test accuracy sharply increases from random to perfect or near-perfect. Figure 3 shows that increasing the training dataset size reduces the number of optimisation steps required before the model can generalise, reducing the steps to grokk. This observation is in line with the original paper that introduced the phenomenon.

It's important to note that Figure 3 shows that the size of the dataset affects the rank that can be used in decomposed learning. This is clear when comparing the decomposed learning test accuracy with rank 12 (blue) with 50%, Figure 3a, and 80%, Figure 3c of the training data. With 50% of the training data, rank 12 starts to generalise later than the baseline and does not reach the same test accuracy with $10^6$ training steps. Whereas with 80% of the training data, rank 12 starts to grok before the baseline. Irrespective of the training dataset size, decomposed learning performs better with higher ranks. In the case of training with 80% of the dataset, Figure 3c, decomposed learning on the token embedding can eradicate the phenomenon. It is also heavily reduced when using 65% of the training data. This result suggests that higher ranks are needed when less representative data is available, and as the dataset becomes more representative, the layer weight matrix can be represented in a low-rank form.

Notably, although decomposed learning on the token embedding reduces the number of steps required for the test accuracy to reach the same accuracy as the train, it does not significantly reduce the number of steps required for the model to reach perfect or near-perfect training accuracy.

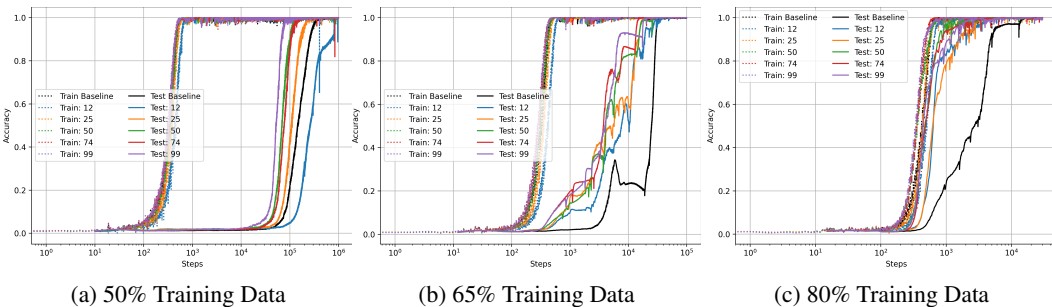

|(a) 50% Training Data|(b) 65% Training Data|(c) 80% Training Data|

Figure 3: Train (dotted) and test (solid) accuracy with decomposed learning on the token embedding layer using ranks 12, 25, 50, 74 and 99, in comparison with the baseline (black) normally trained model.

The observation that higher ranks are needed when less representative data is available is further supported when considering the parameter counts concerning the rank and baseline, Table 1. The table shows that rank 12 and 25 have fewer parameters than the baseline. Therefore, the reduction in steps to grok cannot only be attributed to an increase in the number of parameters, as these ranks reduce the steps to grok in all dataset sizes apart from rank 12 with 50% of the training dataset. This result suggests that optimising the weight matrix, $A$, as $U_k$, $\Sigma_k$ and $V_k^T$ allows for a more efficient form of learning, as the parameter count does not account for reduced steps to grokk.

| Token Embedding $99 \times 128$ | Number of Parameters |
|---|---|
| **Baseline** | $\mathbf{12672} = \underset{99 \times 128}{A}$ |
| Rank 12 | $2868 = \underset{99 \times 12}{U} + \underset{12 \times 12}{\Sigma} + \underset{12 \times 128}{V^T}$ |
| Rank 25 | $6300 = \underset{99 \times 25}{U} + \underset{25 \times 25}{\Sigma} + \underset{25 \times 128}{V^T}$ |
| Rank 50 | $13850 = \underset{99 \times 50}{U} + \underset{50 \times 50}{\Sigma} + \underset{50 \times 128}{V^T}$ |
| Rank 74 | $22274 = \underset{99 \times 74}{U} + \underset{74 \times 74}{\Sigma} + \underset{74 \times 128}{V^T}$ |
| Rank 99 | $32274 = \underset{99 \times 99}{U} + \underset{99 \times 99}{\Sigma} + \underset{99 \times 128}{V^T}$ |

Table 1: Number of parameters for the token embedding layer in the baseline and decomposed learning models at each rank.

## 5.2 POSITION EMBEDDING

The position embedding layer, Figure 4, exhibits a different effect from the token embedding layer. When training on 50% and 65% of the dataset decompose learning takes more steps to grok than the baseline. However, when training on 80% of the dataset, ranks 1 and 2 start to generalise before the baseline. However, it reaches perfect or near-perfect accuracy in approximately the same number of steps as the baseline. Ranks 4 and 5 begin to generalise at the same point as the baseline and reach the training accuracy at a similar number of steps as the baseline. Rank 3 has an initial increase before the baseline, leading to a shallower increase.

Although decomposed learning increased the steps to grok for 50% and 65% of the dataset, it did not increase the number of steps for the model to reach a perfect or near-perfect train accuracy. The models followed the same training accuracy trend as the baseline model. With 80% of the training data, the decomposed learning models increased in training accuracy before the baseline model.

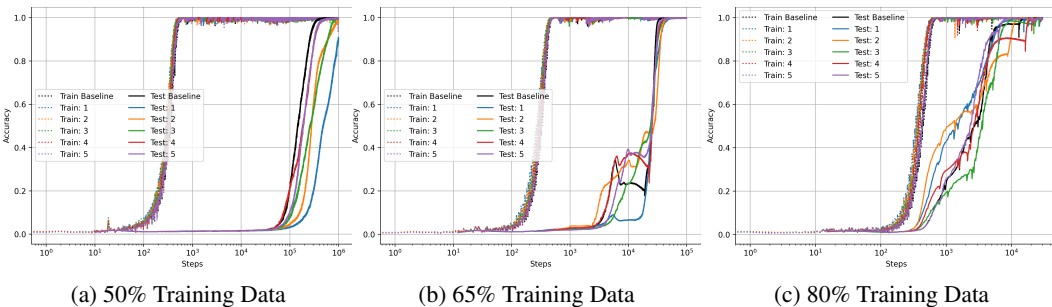

| (a) 50% Training Data | (b) 65% Training Data | (c) 80% Training Data |

Figure 4: Train (dotted) and test (solid) accuracy with decomposed learning on the position embedding layer using ranks 1, 2, 3, 4 and 5, in comparison with the baseline (black) normally trained model .

Table 2 shows the number of parameters associated with each rank representation of the position embedding. Decomposed learning with 65% and 80% of the training dataset approximately reaches the training accuracy at the same time as the baseline. This trend further supports that optimising the weight matrix, $A$, as $U_k$, $\Sigma_k$ and $V_k^T$ allows for a more efficient form of learning as ranks one to four have fewer parameters than the baseline. A potential reason as to why decomposing the position embedding has a marginal effect on the steps to grok could be attributed to the fact that the position embedding has few parameters and is not needed for near-perfect to perfect test accuracy, see Appendix B.1, and thus has little impact of the models learning process.

| Position Embedding $5 \times 128$ | Parameter Size |
|---|---|
| **Baseline** | $640 = \underset{5 \times 128}{A}$ |
| Rank 1 | $134 = \underset{5 \times 1}{U} + \underset{1 \times 1}{\Sigma} + \underset{1 \times 128}{V^T}$ |
| Rank 2 | $270 = \underset{5 \times 2}{U} + \underset{2 \times 2}{\Sigma} + \underset{2 \times 128}{V^T}$ |
| Rank 3 | $408 = \underset{5 \times 3}{U} + \underset{3 \times 3}{\Sigma} + \underset{3 \times 128}{V^T}$ |
| Rank 4 | $548 = \underset{5 \times 4}{U} + \underset{4 \times 4}{\Sigma} + \underset{4 \times 128}{V^T}$ |
| Rank 5 | $690 = \underset{5 \times 5}{U} + \underset{5 \times 5}{\Sigma} + \underset{5 \times 128}{V^T}$ |

Table 2: Number of parameters for the position embedding layer in the baseline and decomposed learning models at each rank.

## 5.3 MULTI-HEAD ATTENTION

Figure 5 shows the effect of decomposed learning on the Multi-Head Attention layer at rank 16, 32, 64, 96 and full rank 128, compared to a normally trained model (baseline). The Multi-Head Attention shows the same general effect as the token embedding layer. Decomposed learning can significantly reduce the number of steps to grokk. Lower ranks can be used as the training dataset increases and still reduce the steps to grokk. For example, rank 16 goes from reaching approximately 20% accuracy with 50% training data, Figure 5a, to approximately 67% with 65% of the training data, Figure 5b. With 80% of the training data at rank 16, Figure 5c, the grokking phenomena does not occur, as the train and test accuracy increase in tandem while reaching near-perfect test accuracy, see Appendix A.1 and Figure 9 for a clearer figure.

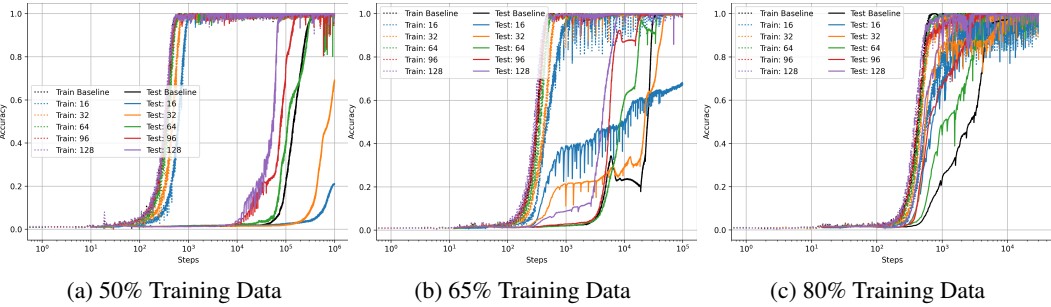

(a) 50% Training Data      (b) 65% Training Data      (c) 80% Training Data

Figure 5: Train (dotted) and test (solid) accuracy with decomposed learning on the multi-head attention layer using ranks 16, 32, 64, 96 and 128, in comparison with the baseline (black) normally trained model.

## 5.4 FEED-FORWARD BLOCKS

Figure 6 shows the effect of decomposed learning on the feed-forward blocks at rank 16, 32, 64, 96 and full rank 128, compared to the normally trained model (baseline). The Feed-forward blocks show the same effect as the token embedding layer and multi-head attention, although less dramatic. Increasing the dataset size allows for lower ranks to be used. This effect is demonstrated with rank 16 with a 50% training dataset, resulting in approximately a 50% test accuracy, but with 65% and 80% training data reaching 95%+ test accuracy. It also shows that decomposed learning can significantly reduce the number of steps to grokk.

Using a rank of 16 for the feed-forward blocks always increases the steps required to grokk. This effect could be because it has approximately 84% fewer parameters than the baseline, thus reducing the representational freedom. However, rank 32 has 67% fewer parameters than the baseline and can grok in a similar number of steps as the baseline with 65% and 80% training data. Therefore, why rank 16 performs worse than the baseline needs further investigation, which may elucidate how the number of parameters affects the performance. See Appendix B.3 for further investigation.

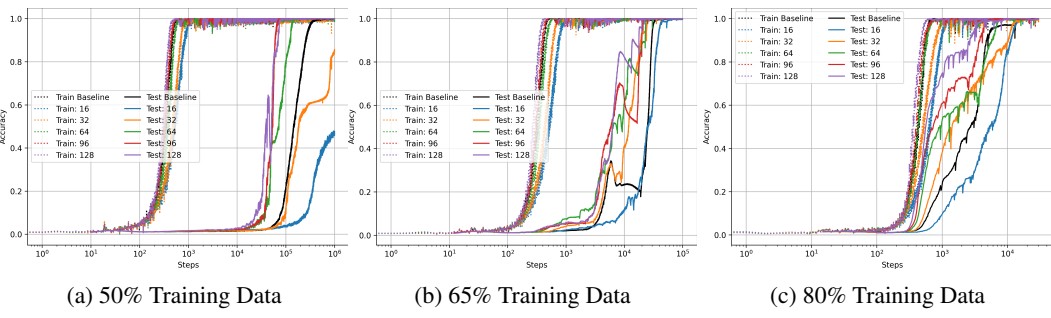

(a) 50% Training Data      (b) 65% Training Data      (c) 80% Training Data

Figure 6: Train (dotted) and test (solid) accuracy with decomposed learning on the feed-forward blocks using ranks 16, 32, 64, 96 and 128, in comparison with the baseline normally trained model (black).

## 5.5 OUTPUT LAYER

The Output layer, Figure 7, exhibits the same effect that decomposed learning can reduce the steps to grokk. As noted with the token embedding layer, multi-head attention and the feed-forwards blocks, this effect is most apparent with higher ranks. The impact of reducing grokking is less dramatic on the output layer.

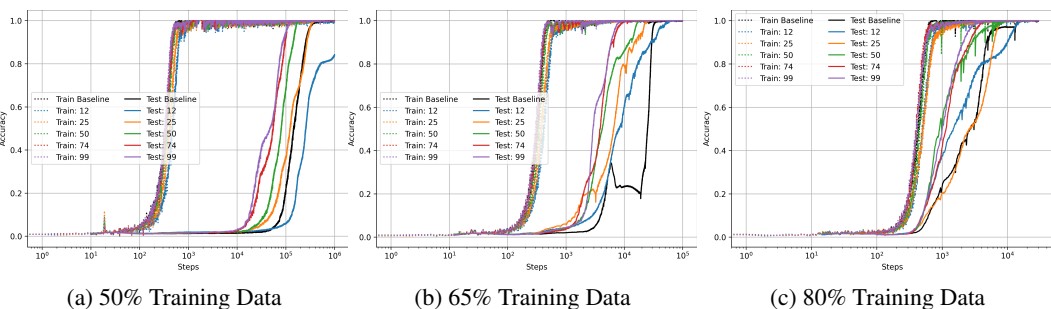

(a) 50% Training Data     (b) 65% Training Data     (c) 80% Training Data

Figure 7: Train (dotted) and test (solid) accuracy with decomposed learning on the output layer using ranks 12, 25, 50, 74 and 99, in comparison with the baseline normally trained model (black).

### 5.6 TOKEN EMBEDDING, MULTI-HEAD ATTENTION FEED-FORWARD BLOCKS AND OUTPUT LAYER

The token embedding, multi-head attention, feed-forward blocks and the output layer reduced the steps to grok. In some cases, it removed the phenomenon's occurrence entirely. Therefore, to better understand the effect of decomposed learning and the rank. These layers are decomposed to 12.5%, 25%, 50%, 75% and 100%, of the full rank available for the respective layers and trained, see Figure 8. The figure shows that decomposed learning can reduce and completely avoid the grokking phenomenon.

With 50% training data, Figure 8a, and 100%, 75% and 50% decompositions can reach the perfect or near-perfect test accuracy before the baseline. However, 25% and 12.5% decompositions start to generalise but cannot outperform the baseline model or reach a perfect or near-perfect test accuracy. For 100% decomposition, the model starts to generalise (achieve above random accuracy) before the model has completely memorised the training dataset; see Appendix A.2 Figure 10 for a more explicit figure. The 100% decomposition also requires 61.67 times fewer steps to reach a 1% generalisation gap; see Appendix A.2 Figure 10. Although 75% decomposition reaches perfect test accuracy at step $2.117 \times 10^4$, approximately at step $3 \times 10^5$, the accuracy suddenly drops to approximately 82%. This change in test accuracy indicates that the learning rate is too high to continue to optimise the model.

With 65% and 80% training data, Figures 8b and 8c, the grokking phenomena is removed, with an improvement in training accuracy corresponding with an improvement in test accuracy. This is most dramatic with 80% of the training data, Figure 8c, where the models train and test accuracy increase in tandem, as observed in conventional training setups. It's noted that the training accuracy now takes more steps than the baseline to achieve perfect or near-perfect performance. However, perfect or near-perfect test accuracy is achieved before the baseline. In addition, Figure 8c, using 75% of the ranks, results in a model that reaches near-perfect accuracy before the 100% decomposition; exactly why this is unclear and needs further exploration.

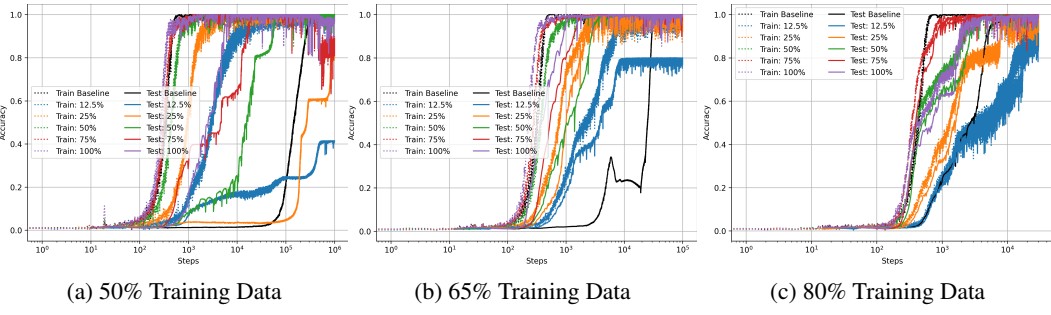

(a) 50% Training Data     (b) 65% Training Data     (c) 80% Training Data

Figure 8: Train (dotted) and test (solid) accuracy with decomposed learning on the token embedding, multi-head attention, feed-forward blocks and output layer using 12.5%, 25%, 50%, 75% and 100% of the ranks in comparison with the baseline (black) normally trained model.

## 6 Discussion

**Decomposed Learning Encourages the Learning of More Generalisable Representations** This method can prevent grokking even with fewer parameters (as seen in the token embedding at rank 25). This structured initialisation, with orthonormality and diagonalization of the matrices that are not maintained through training, introduces an implicit regularisation effect, that encourages the reduction of the stable rank through training more rapidly, see Appendix D, which allows the model to generalise faster when a sufficiently high rank is used.

**More Data Fewer Ranks** Our results show that within this setup, fewer ranks are required for the model to generalise as the dataset increases and becomes more representative of the learning problem. Further highlighting the need for extensive and representative datasets for efficient deep-learning training. This suggests that when finetuning with methods such as LoRA, if the data does not adequately represent the solution space, more ranks should be used; however, if the data does, then fewer ranks can be used.

**Different Layers Can Learn With Varying Levels of Rank Reduction While Maintaining Performance and Mitigating or Preventing Grokking** Our results show that different layers require more ranks than other layers to maintain performance and reduce or avoid grokking specifically, in order of the layers that require the fewest ranks to maintain performance, the token embedding, output layer, feed-forward blocks, multi-head attention and position embedding. However, precisely what this means regarding the learning process of neural networks is unclear and requires further study.

**Free Lunch for Compression** These results show that often using the full-rank representation of the weight matrix in $U\Sigma V^T$ results in a model that can avoid and mitigate the grokking phenomena the most, especially in reduced data regimes such as 50% training data. This, therefore, allows for a form of pre-training compression, where using decomposed learning, one can specify the size of the model required for inference, then train in the full decomposed rank representation to allow the model to achieve better performance and then recompose this matrix at the end, such that the inference cost does not change, see Figure 1. Of course, if the model is trained with less than $k$ ranks from equation 5, then the model can naturally be compressed after being recomposed into $A$ and subsequently decomposed with SVD to the rank selected without harming the performance.

## 7 Conclusion

In this paper, we have explored the grokking phenomenon in neural networks, the behaviour where models generalise to the test dataset significantly after overfitting the training dataset. We proposed a novel method rooted in linear algebra to mitigate or eliminate this issue. By using SVD to change the representation of the weight matrix, $A$, into the product of three matrices $U$, $\Sigma$ and $V^T$ and training on this representation.

Our results show that this method reduces the complexity of the learning task, allowing the model to generalise more effectively with fewer parameters, especially as the dataset becomes more representative. Moreover, the flexibility of applying varying levels of rank reduction across different layers shows promise in maintaining performance while avoiding the grokking phenomenon.

These findings further contribute to the growing body of work demonstrating that grokking is an artefact of poor training setups instead of being a natural or regularly occurring phenomenon within deep learning. They also open up new possibilities for more efficient training and inference of neural networks, primarily in transformer-based models, where parameter efficiency and generalisation are critical. This novel approach contributes to a deeper understanding of grokking and offers a practical solution for overcoming its challenges in machine learning.

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

# A ENLARGED FIGURES

## A.1 MULTI-HEAD ATTENTION AT RANK 16

Decomposed learning on the Multi-Head Attention at Rank 16 with 80% training data is shown in Figure 9. This figure shows that the train (blue) and test (orange) accuracy increase in tandem for decomposed learning, reaching a near-perfect test accuracy at the end of training.

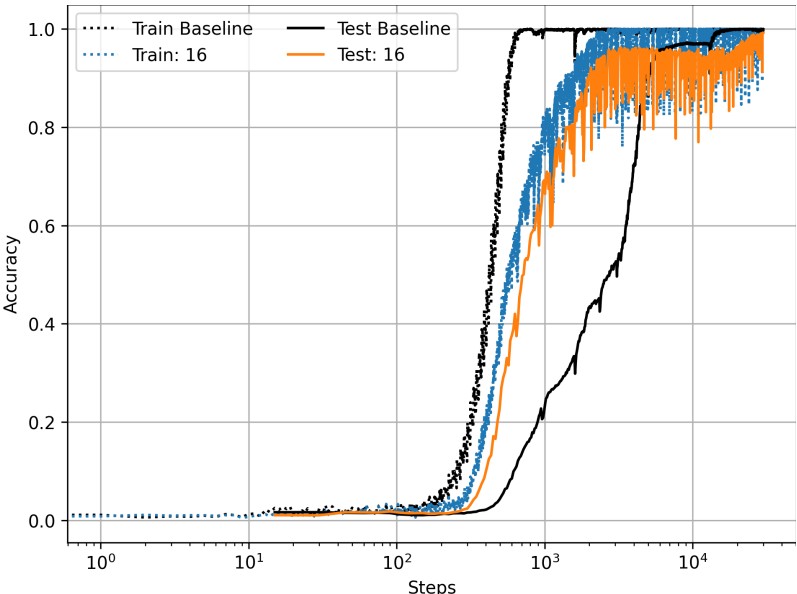

Figure 9: Train (dotted) and test (solid) accuracy with decomposed learning on the multi-head attention at rank 16 in comparison with the baseline (black) normally trained model with 80% training data.

## A.2 TOKEN EMBEDDING, MULTI-HEAD ATTENTION FEED-FORWARD BLOCKS AND OUTPUT LAYER: 100% DECOMPOSED

Decomposed learning on the token embedding, multi-head attention, feed-forward blocks and output layer using 100% of the ranks and 50% training data is shown in Figure 10. This figure shows that the 100% decomposed learning requires 61.67 times fewer steps to reach a 1% generalisation gap (stars) when compared to the baseline model. Also of note is that the model starts to generalise before the model has fully overfit the training dataset.

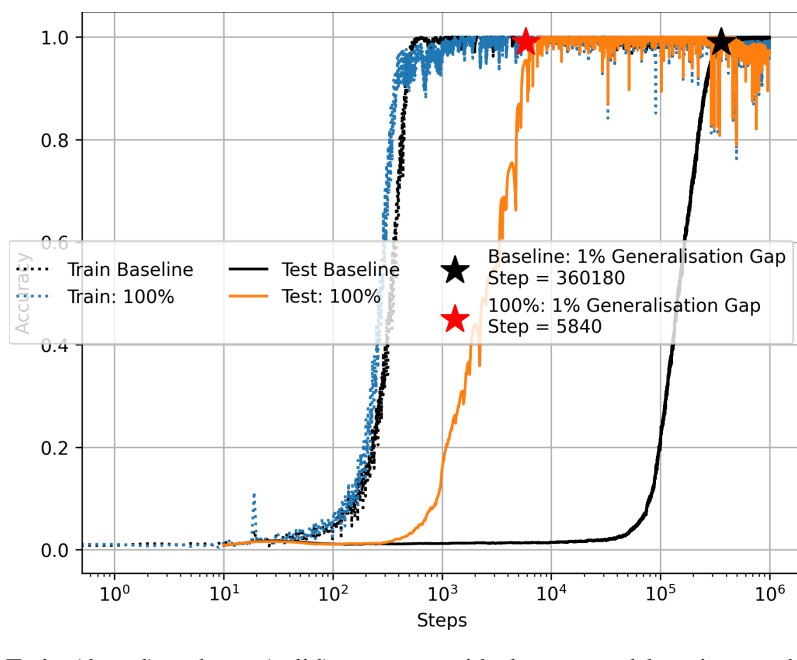

Figure 10: Train (dotted) and test (solid) accuracy with decomposed learning on the token embedding, multi-head attention, feed-forward blocks and output layer using 100% of the ranks in comparison with the baseline (black) normally trained model.

## B ADDITIONAL LAYER ANALYSIS

This section further explores the effect of different layers within the grokking setups. Section B.1 explores the impact of the position embedding and its influence on the model's performance. Section B.2 explores how decomposed learning on the QKV_in and Proj_out independently affects the learning and grokking phenomenon. Finally, section B.3 explores how decomposed learning on the fully connected layers FC1 and FC2 independently affects the grokking phenomenon.

### B.1 THE IMPACT OF THE POSITION EMBEDDING

The effect of learning without the position embedding is shown in Figure 11. This figure shows the model can still achieve perfect or near-perfect test accuracy within the set number of optimisation steps. This demonstrates that the position embedding has little impact on the grokking phenomenon or the model's generalisation ability.

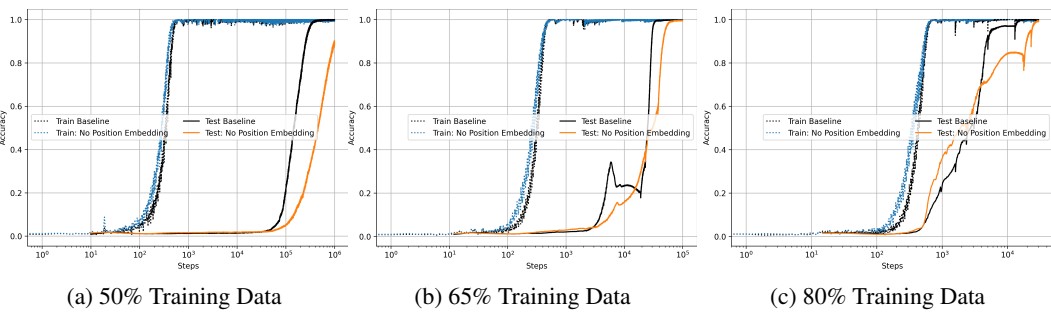

(a) 50% Training Data     (b) 65% Training Data     (c) 80% Training Data

Figure 11: Train (dotted) and test (solid) accuracy of a model trained without the position embedding, in comparison with the baseline normally trained model (black).

## B.2 MULTI-HEAD ATTENTION

The multi-head attention layer comprises two linear layers, QKV_in and Proj_out. QKV_in is the query, key and value layers represented as one linear layer with an input of 128 and an output of 384. The Proj_out layer has an input of 128 and an output of 128.

Decomposed learning on the QKV_in layer, Figure 12, the general trend is that the more ranks used with decomposed learning, the fewer steps required for effective generalisation, with rank 128 being the best performing regardless of training data amounts. In addition, fewer ranks still perform well with rank 16, being able to generalise just after the baseline with 65% of the training data, Figure 12b and before with 80% of training data Figure 12c.

Exploring the effect of decomposed learning on the Proj_out layer, Figure 13, demonstrates a slightly different effect from the QKV_in layer regarding rank 128 with 50% training data, Figure 13a. This figure shows that decomposed learning in a full-rank form takes slightly longer to generalise than the baseline. Although when training with 65%, Figure 13b and 80% training data, Figure 13c, rank 128 generalises before the baseline. It is of note that with 50% training at rank 16, Figure 13a, the Proj_out layer is unable to generalise effectively, which continues, however not as dramatic, till training with 80% training data, Figure 13c, where it reaches near-perfect test accuracy. Overall, the effect of decomposed learning on the Proj_out layer is marginal.

Decomposed learning on the QKV_in and Proj_out layers independently supports the finding that training with more data means fewer ranks are required.

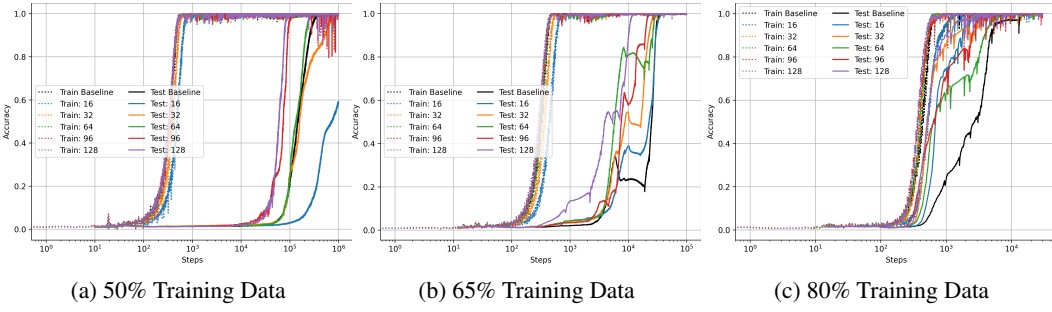

(a) 50% Training Data      (b) 65% Training Data      (c) 80% Training Data

Figure 12: Train (dotted) and test (solid) accuracy with decomposed learning on the multi-head attention QKV_in linear layer using ranks 16, 32, 64, 96 and 128, in comparison with the baseline (black) normally trained model.

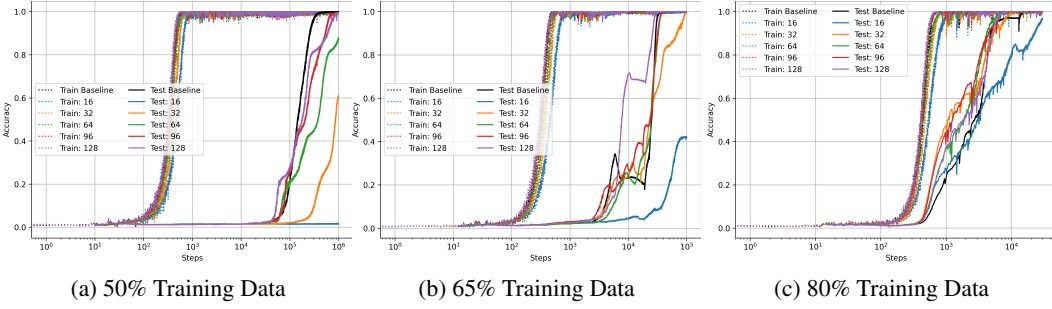

(a) 50% Training Data      (b) 65% Training Data      (c) 80% Training Data

Figure 13: Train (dotted) and test (solid) accuracy with decomposed learning on the multi-head attention Proj_out linear layer using ranks 16, 32, 64, 96 and 128, in comparison with the baseline (black) normally trained model.

### B.3 FEED-FORWARD BLOCKS

The feed-forward blocks comprise two fully connected layers and a GeLU activation function (Hendrycks & Gimpel, 2023): The first fully connected layer, FC1, has 128 inputs with 512 outputs, and the second, FC2, has 512 inputs with 128 outputs.

The effect of decomposed learning on FC1 is shown in Figure 14. This figure clearly shows the role of data and the number of ranks that can be used for decomposed learning, with rank 12 only starting to generalise at the end of training with 50% of the training data, Figure 14a, but nearing complete generalisation with 65% training data, Figure 14b, and fully generalising with 80% training data, Figure 14c. Conversely, with FC2, Figure 15, this effect is still present although less dramatic, with rank 12 being able to generalise with 50% and 65% of the training data albeit after the baseline, Figure 15a and 15b and 80% of the training data resulting in generalising just before 15c. For FC1, the best performing rank with 50% of the data is the full rank representation, rank 128; however, for FC2, this is rank 96.

The different effects of rank 12 on FC1 and FC2 suggest that the structure of the layer plays an essential role in what rank it can represent. FC1 can be viewed as projecting a lower dimension into a higher dimension, and FC2 can be viewed as projecting a higher dimension into a smaller dimension. These results suggest that when going from lower to higher, more ranks are required, and when going from higher to lower, fewer ranks are required.

These results indicated that rank 16 performs worse than the baseline when decomposing FC1 and FC2 together because FC1 is so strongly affected by rank 16 that even though FC2 is hardly affected, the negative impact of FC1 is too strong, resulting in worse performance than the baseline.

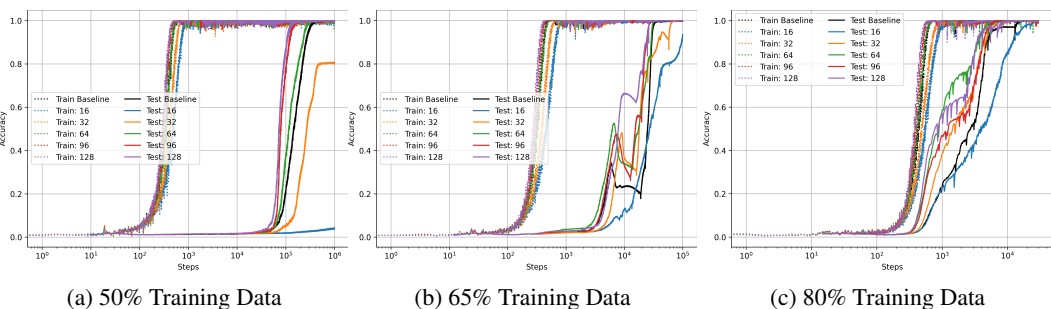

(a) 50% Training Data     (b) 65% Training Data     (c) 80% Training Data

Figure 14: Train (dotted) and test (solid) accuracy with decomposed learning on the FC1 layer using ranks 16, 32, 64, 96 and 128, in comparison with the baseline normally trained model (black).

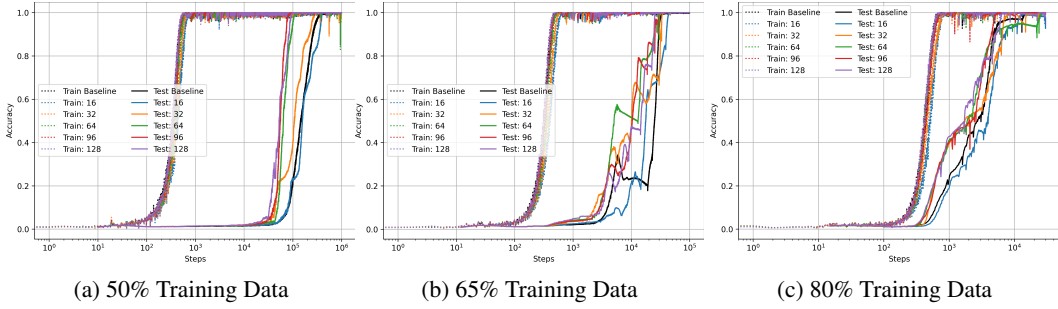

(a) 50% Training Data     (b) 65% Training Data     (c) 80% Training Data

Figure 15: Train (dotted) and test (solid) accuracy with decomposed learning on the FC2 layer using ranks 16, 32, 64, 96 and 128, in comparison with the baseline normally trained model (black).

# C $U_k$, $\Sigma_k$ AND $V_k^T$ AFTER TRAINING

A matrix, $A$, is orthogonal if and only if $AA^T = A^T A = I$, performing SVD on a matrix, $M$, returns $U_k \Sigma_k V_k^T$, where $U_k$ and $V_k$ are orthogonal matrices and $\Sigma$ is diagonal and $k$ is the rank. In decomposed learning, the orthogonality of $U_k$ and $V_k$ is not maintained, allowing $U_k$ and $V_k$ to become non-orthogonal matrices, Figures 16 and 17 show that $U_k$ and $V_k$ are non-orthogonal after training in decomposed learning on the token embedding for ranks 12, 25, 50, 75 and 99 with 50% of the training data.

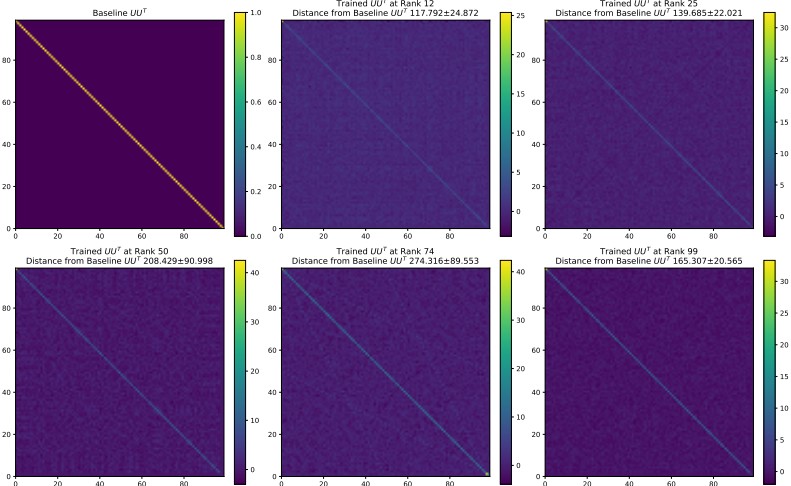

Figure 16: The $UU^T$ of the Token Embedding after training in the Base (conventionally trained) and decomposed with rank 12, 25, 50, 74 and 99, respectively. Results averaged from 5 models.

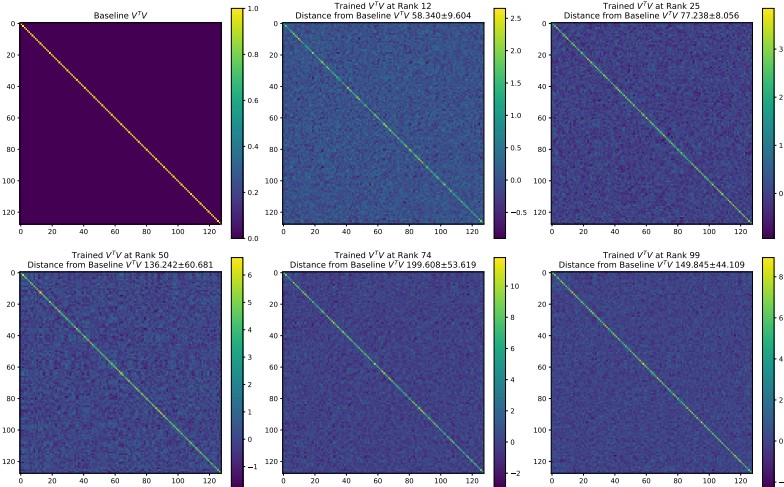

Figure 17: The $V^T V$ of the Token Embedding after training in the Base (conventionally trained) and decomposed with rank 12, 25, 50, 74 and 99, respectively. Results averaged from 5 models.

In decomposed learning, the diagonality of $\Sigma_k$ is not maintained and can become non-diagonal. Figure 18 shows how non-diagonal $\Sigma_k$ is after training in decomposed learning on the token embedding for ranks 12, 25, 50, 75 and 99. It is interesting to note that even though $\Sigma_k$ becomes non-diagonal, the diagonal does have significantly higher values than the rest of the matrix.

Although $U_k$ and $V_k$ become non-orthogonal matrices and $\Sigma_k$ becomes non-diagonal during training, when $U_k$, $\Sigma_k$ and $V_k^T$ are recomposed into $M$, the rank of $M$ is the same rank as used in

decomposed learning, as shown in Figure 19 with the diagonal becoming zero after the specified rank. Allowing for a low rank representation to be used for inference. Interestingly, the token embedding trained in the decomposed learning has top singular values ten times larger than the baseline model.

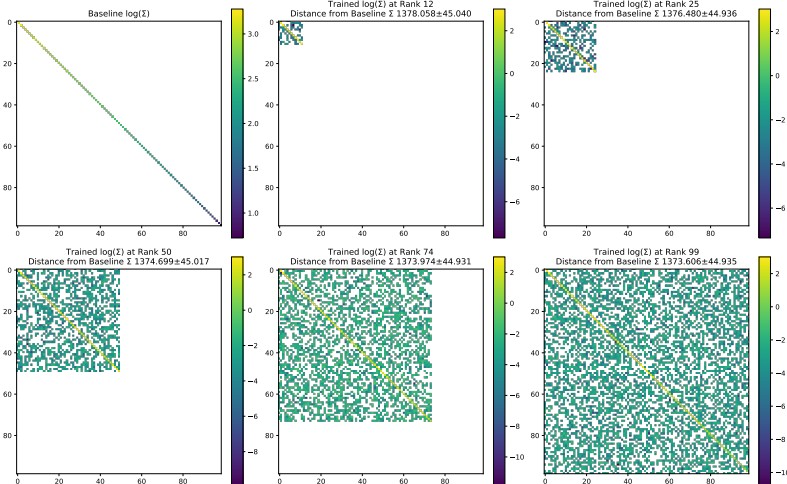

Figure 18: The log singular values ($\log \Sigma$) of the Token Embedding after training in the Base (conventionally trained) and decomposed with rank 12, 25, 50, 74 and 99, respectively. White Pixels are zero. Results averaged from 5 models.

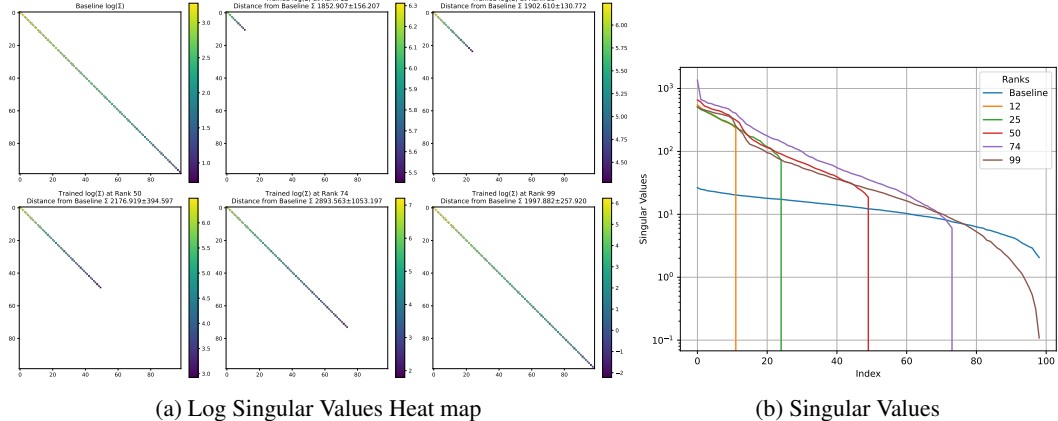

(a) Log Singular Values Heat map        (b) Singular Values

Figure 19: (a) The log singular values ($\log \Sigma$) and (b) the singular values of the Token Embedding after training in the Base (conventionally trained) and decomposed with rank 12, 25, 50, 74 and 99, respectively and subsequently reconstructed. White Pixels in (a) are zero. Results averaged from 5 models.

Although these figures display results for the token embedding, the general results that $U_k$ and $V_k$ become non-orthogonal matrices and $\Sigma_k$ becomes non-diagonal during training, and when $U_k, \Sigma_k, V_k^T$ are recomposed into $M$, the rank of $M$ is the same rank used in decomposed learning applies to all layers where decomposed learning occurs.

## D    SPECTRAL ANALYSIS THROUGH TRAINING

This section explores how the layers' normalised stable rank (Rudelson & Vershynin, 2007) changes through training. To calculate the normalised stable rank of the decomposed layer, $U_k$, $\Sigma_k$ and $V_k$ are recomposed ($U_k \Sigma_k V_k^T$) to create the matrix, $A$, which then is analysed.

The stable rank equation 6 is the Frobenius norm squared divided by the squared spectral norm of the matrix; it is normalised by dividing the result by the full rank, $min(m, n)$, where $m$ is the rows, and $n$ is the columns of $A$.

$$Stable\ Rank = (\frac{||A||_F^2}{||A||_2^2}) \tag{6}$$

The effect of decomposed learning, reducing the number of steps required before reaching near-perfect or perfect accuracy, is most dramatic when all layers except the position embedding are decomposed. Therefore, we explore how the stable ranks of layers change through training, with 50% of the training data, and decompose learning on all layers except the position embedding by analysing the stable rank and the accuracy at 200 points through training, Figure 20.

For the baseline (normally trained model), top left in Figure 20, there is a slow transition from a high stable rank to a low stable rank throughout training. As the stable rank decreases, the test accuracy of the model increases. For decomposed learning with 100% of the ranks for all layers except for the position embedding, top centre in Figure 20, the start of training follows the baseline, with a slow progression from high stable rank to low stable rank and then at approximately point 50, the stable ranks of the layers dramatically reduce close to zero which corresponds with a dramatic increase in accuracy and the model achieving perfect or near-perfect test accuracy. This finding follows for decomposing learning with 75% and 50% of the ranks for all layers except for the position embedding. However, for decomposed learning with 25% and 12.5% of the ranks for all layers except for the position embedding, the model makes the same transition from high to low, stable rank through training across layers, albeit not as high initially and achieves a similar but slightly lower stable rank at the end of training, with this the 25% case generalises at approximately the same time as the baseline model. In contrast, the 12.5% case cannot generalise to the same level as the baseline.

Observing the difference between the baseline and decomposed learning, Figure 21, on 25% and 12.5% of the ranks for all layers except for the position embedding, the baseline starts with a higher stable rank than the decomposed learning on 25% and 12.5%. This indicates that it is **not** simply getting a low, stable rank that is important for generalisation. Instead, transitioning from a sufficiently high to low, stable rank is important for generalisation, or at least an indicator.

These results provide an explanation as to why decomposed learning works. It shows that decomposed learning can speed up the process of transitioning from a sufficiently high, stable rank to a low, stable rank if a high enough initial rank is used, which is dependent on the amount of data. This suggests that it helps the implicit regularisation process in learning to reduce the stable rank more effectively and thus can reduce the steps required for grokking.

This general finding extends when exploring decomposed learning with 100% of the ranks for all layers except for the position embedding with 65%, Figure 22, and 80%, Figure 23, of the training data. These results suggest that learning can start from a lower-dimensional manifold as the data becomes more representative of the solution space. In the context of the current grokking theory, this analysis of the stable rank suggests that lazy (linear) and rich (feature) learning as proposed by Kumar et al. (2024) happen in different dimensions with lazy (linear) learning happening in higher dimensions and rich (feature) learning happening in lower dimensions.

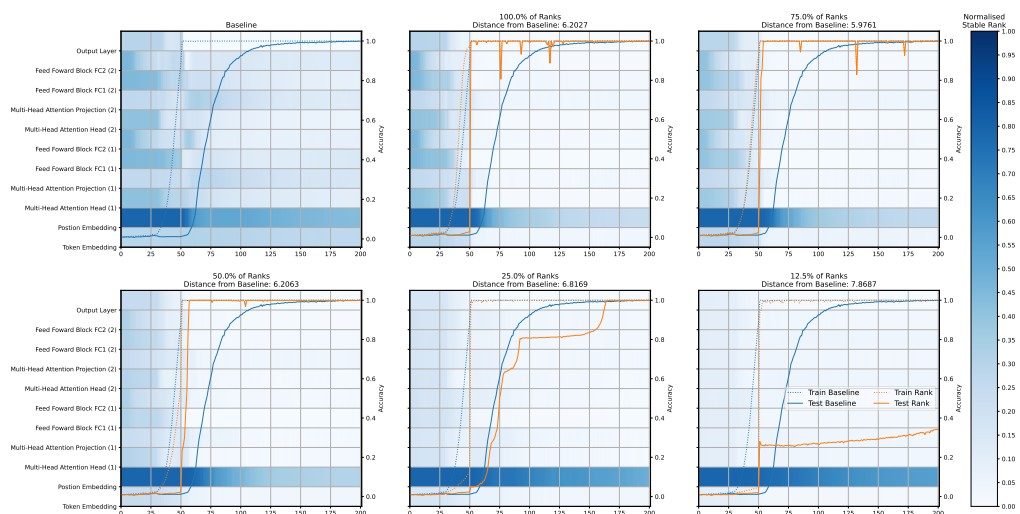

Figure 20: The normalised stable ranks of layers through training with 50% of the training data for the baseline and the decomposed learning on all layers except the position embedding at 100%, 75%, 50%, 25% and 12.5% of full rank for the respective layers. The distance from the baseline is the Euclidean distance between baseline stable ranks and the decomposed learning stable through training for all layers. The train and test accuracy of the baseline model is plotted in blue, and the train and test accuracy of the decomposed model is plotted in orange. The x-axis is not linear, and the distance between steps varies. The mean from 5 runs is reported.

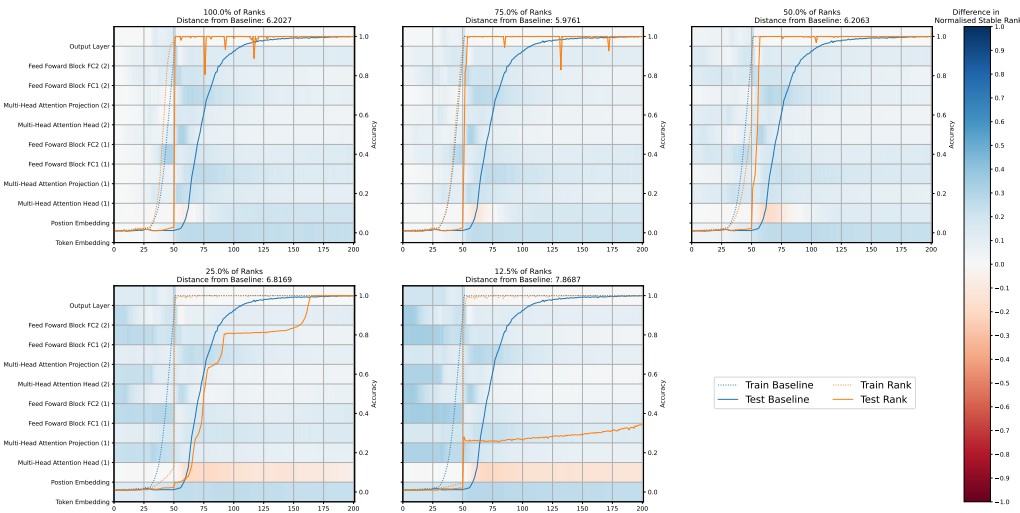

Figure 21: The difference between the baseline and the decomposed model normalised stable ranks through training with 50% of the training data. Blue indicates the baseline model has a higher stable rank, whiteish cells indicate little difference between stable ranks, and red cells indicate the decomposed model has a higher stable rank. The distance from the baseline is the Euclidean distance between baseline stable ranks and the decomposed learning stable through training for all layers. The train and test accuracy of the baseline model is plotted in blue, and the train and test accuracy of the decomposed model is plotted in orange. All layers except the position embedding were decomposed at 100% (top left), 75%, 50%, 25% and 12.5% of the full rank for each respective layer. The x-axis is not linear, and the distance between steps varies. The mean from 5 runs is reported.

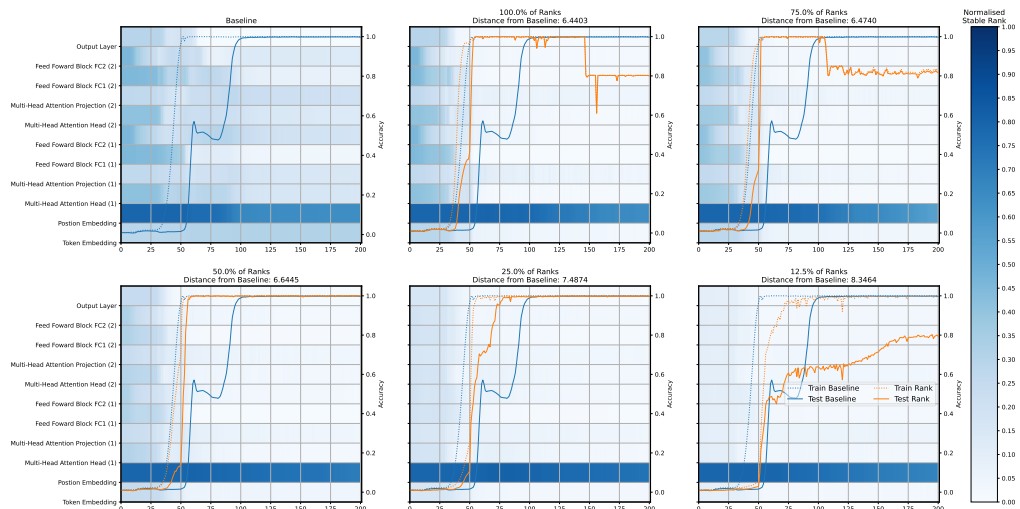

Figure 22: The normalised stable ranks of layers through training with 65% of the training data for the baseline and the decomposed learning on all layers except the position embedding at 100%, 75%, 50%, 25% and 12.5% of full rank for the respective layers. The distance from the baseline is the Euclidean distance between baseline stable ranks and the decomposed learning stable through training for all layers. The train and test accuracy of the baseline model is plotted in blue, and the train and test accuracy of the decomposed model is plotted in orange. The x-axis is not linear, and the distance between steps varies. The mean from 5 runs is reported.

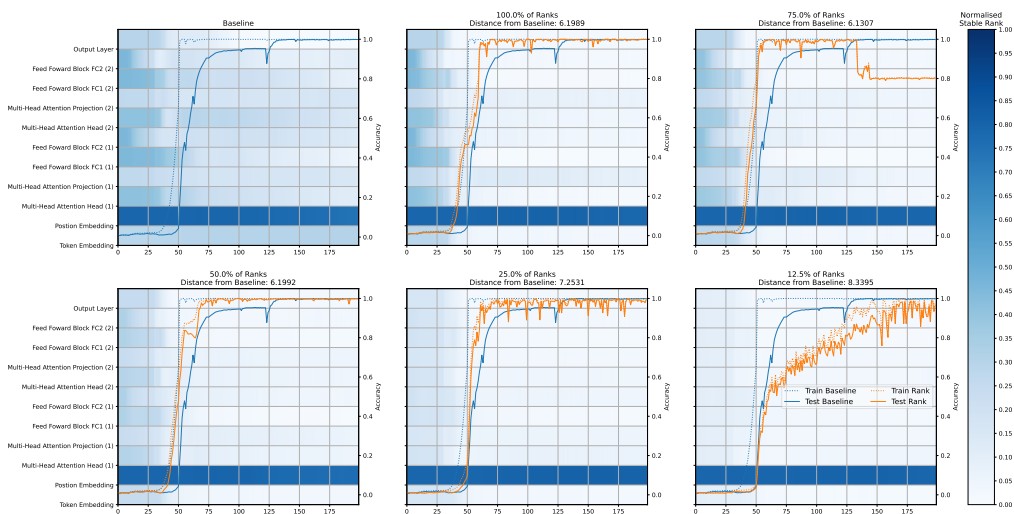

Figure 23: The normalised stable ranks of layers through training with 80% of the training data for the baseline and the decomposed learning on all layers except the position embedding at 100%, 75%, 50%, 25% and 12.5% of full rank for the respective layers. The distance from the baseline is the Euclidean distance between baseline stable ranks and the decomposed learning stable through training for all layers. The train and test accuracy of the baseline model is plotted in blue, and the train and test accuracy of the decomposed model is plotted in orange. The x-axis is not linear, and the distance between steps varies. The mean from 5 runs is reported.

# E   THE EFFECT OF WEIGHT DECAY ON DECOMPOSED LEARNING

Weight decay, or $L_2$ regularisation, is an explicit regularisation technique that penalises large weights and encourages small weights within the neural networks (Goodfellow et al., 2016). To explore the effect of weight decay on decomposed learning, we trained an MLP network with layer sizes 784-256-10, i.e. a network with one hidden layer of width 256, on MNIST for 25 epochs using the SGD optimiser with a learning rate, momentum, and batch size of 0.001, 0.9, and 256, respectively. The model was then trained with a weight decay of 0, 1.0, 0.1, 0.01, 0.001, and 0.0001 and the highest test accuracy achieved during training was recorded from the Baseline. Each layer was decomposed independently MLP 1, Figure 24a, MLP 2, Figure 24b, MLP 3, Figure 24c and then all layers where decomposed simultaneously, Figure 24d, from 100% to 10% of the full rank with 10% jumps the highest test accuracy achieved during training was recorded.

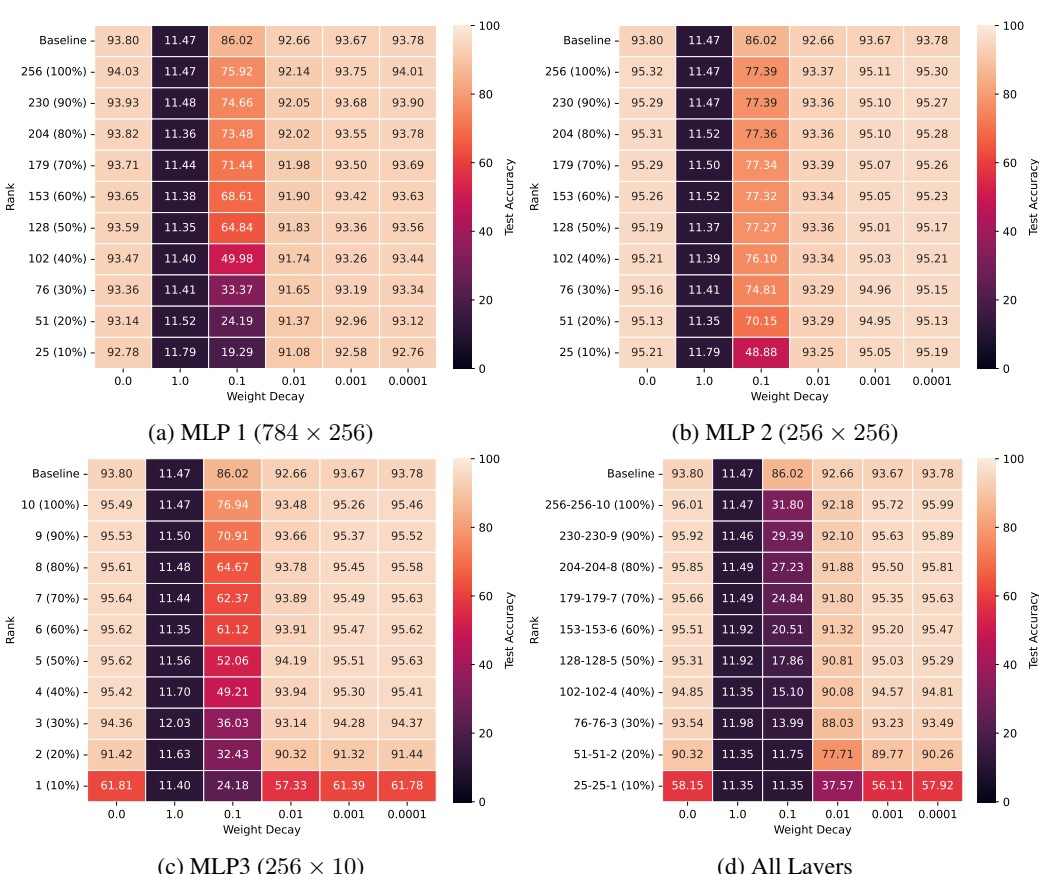

(a) MLP 1 (784 × 256)          (b) MLP 2 (256 × 256)

(c) MLP3 (256 × 10)          (d) All Layers

Figure 24: The effect of weight decay on test accuracy and the rank that can be used in decomposed learning for each layer (a,b,c) and all layers (d). The Baseline is a model trained conventionally without decomposed learning. Average highest test accuracy from 10 models.

Figure 24 shows that the Baseline model, trained without decomposed learning, performs better without weight decay (93.80%). With a weight decay of 1.0, the model achieved just above random accuracy, 11.47%. However, it should be noted that a weight decay of 1 is very large. As the weight decay decreases, the model performance gets closer to training without weight decay, with accuracies of 86.02%, 92.66%, 93.67% and 93.78% for weight decay values of 0.1, 0.01, 0.001 and 0.0001 respectively.

For decomposed learning on MLP 1, Figure 24a, weight decay prevents the method from performing as well with lower ranks. When not using weight decay, 0.0, the layer can be decomposed to 80% before achieving a lower accuracy than the Baseline. However, when using a weight decay of 0.1 or 0.01, no form of decomposed learning can outperform the baseline performance using that weight

decay. Note for weight decay of 0.1, decreasing the number ranks dramatically affected the test accuracy, which was not observed when using weight decay of zero. For weight decay of 0.001 and 0.0001, only a decomposed learning with 90% and 80% of the ranks could be used before, resulting in a worse test accuracy than the Baseline. The same trend that the weight decay reduces the number of ranks that can be used in decomposed learning while still maintaining or improving test accuracy, with larger values having a more substantial effect, is also observed in MLP 2, Figure 24b, and MLP 3, Figure 24c.

For the all-layer case, Figure 24d, the effect of weight decay is more apparent and has an even more significant impact at lower ranks. Without weight decay, decomposed learning can use 40% of each layer's ranks while still achieving better test accuracy of 94.85%, resulting in a training and inference model 67.1% and 59.37% of the size of the baseline model, respectively. With a weight decay of 0.1, decomposed learning at any rank results in a model that performs dramatically worse even at full rank decomposition, which without weight decay results in a model 2.21% increase in test accuracy but with weight decay of 0.01 a 54.22% decrease in test accuracy.

These results show that if using decomposed learning, weight decay should be avoided for the layer being decomposed to enable fewer ranks and, thus, a higher compression ratio during training and inference while maintaining or improving performance. This result aligns with recent work by Yunis et al. (2024), which shows that weight decay also encourages rank minimization. Therefore, it can easily be envisioned, given that decomposed learning also encourages rank minimization, Appendix D, that when both are applied, the regularisation effect is too strong and thus degrades performance.

# F    GROKKING INDUCED MNIST

We explore the effect of decomposed learning on the grokking-induced MNIST (Liu et al., 2023). We follow the same setup in Liu et al. (2023) and train an MLP network with layer sizes 784-200-10, i.e. a network with one hidden layer of width 200, on the MNIST dataset using 1000 data points and the MSE loss. The model is optimised with the AdamW using a learning rate of 0.001 and weight decay of 0.01, a batch size of 200, as described by Liu et al. (2023). The model is initialised with all parameters multiplied by 8 to increase the weight norm of the layers.

The first layer is decomposed to 100%, 75%, 50%, 25% and 12.5% of the total ranks and the train and test accuracy and loss are recorded throughout training, Figure 25. When decomposing on the first layer, the model cannot reach the same test accuracy as the baseline regardless of the rank; interestingly, using 50% or fewer ranks allows the model to perform better than using more ranks but still lower than the baseline. We attribute this occurrence to weight decay, as Appendix E shows that decomposed learning performs poorly with weight decay. Grokking was not observed for all ranks, with the train and test accuracy increasing in tandem.

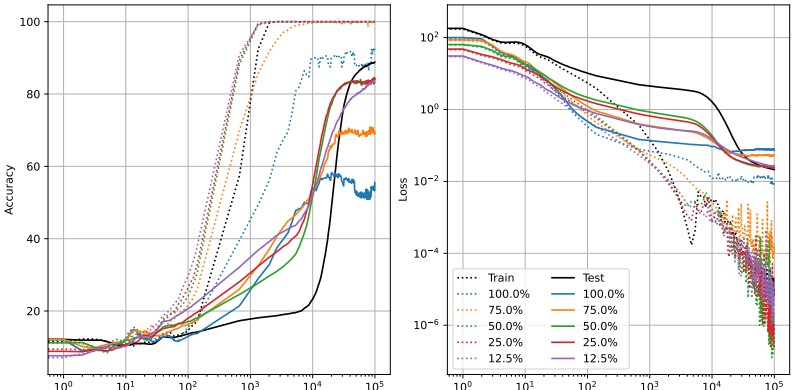

Figure 25: Train (dotted) and test (solid) accuracy with decomposed learning on Layer 1 using 12.5%, 25%, 50%, 75% and 100% of the ranks in comparison with the baseline normally trained model (black). Mean from 3 runs.

The second layer is decomposed to 100%, 75%, 50%, 25% and 12.5% of the total ranks and the train and test accuracy and loss are recorded throughout training, Figure 26. When decomposing on the second layer, the model outperforms the baseline regardless of the rank. For 100%, the train and test accuracy increases in tandem and mitigates the grokking phenomena.

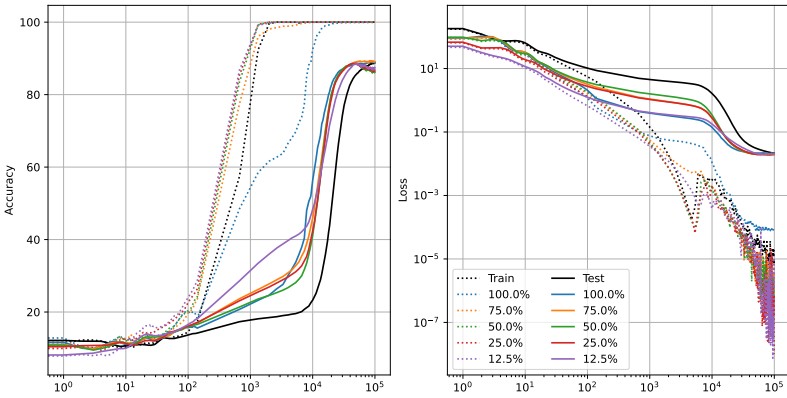

Figure 26: Train (dotted) and test (solid) accuracy (left) and loss (right) with decomposed learning on Layer 2 using 12.5%, 25%, 50%, 75% and 100% of the ranks in comparison with the baseline normally trained model (black). Mean from 3 runs.

The third layer is decomposed to 100%, 75%, 50%, 25% and 12.5% of the total ranks and the train and test accuracy and loss are recorded throughout training, Figure 27. When decomposing on the third layer, the model outperforms the baseline with 100% of the ranks, but all other decompositions result in worse performance. For 75% and below, the training accuracy peaks a lot higher than the test accuracy and then returns to slightly above the test accuracy towards the end of the training.

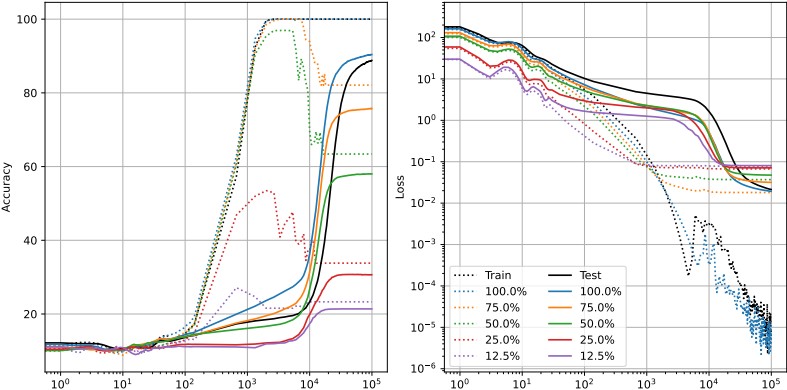

Figure 27: Train (dotted) and test (solid) accuracy with decomposed learning on Layer 3 using 12.5%, 25%, 50%, 75% and 100% of the ranks in comparison with the baseline normally trained model (black). Mean from 3 runs.

As decomposed learning on the second and third layer with 100% resulted in models that improved performance, decomposed learning was performed on both layers at 100% simultaneously, and the train and test accuracy and loss were recorded throughout training, Figure 28. When both layers decomposed at 100% simultaneously, the grokking phenomena is effectively mitigated, as the train and test accuracy increase in tandem without delayed generalisation and decomposed learning results in a model with higher test accuracy.

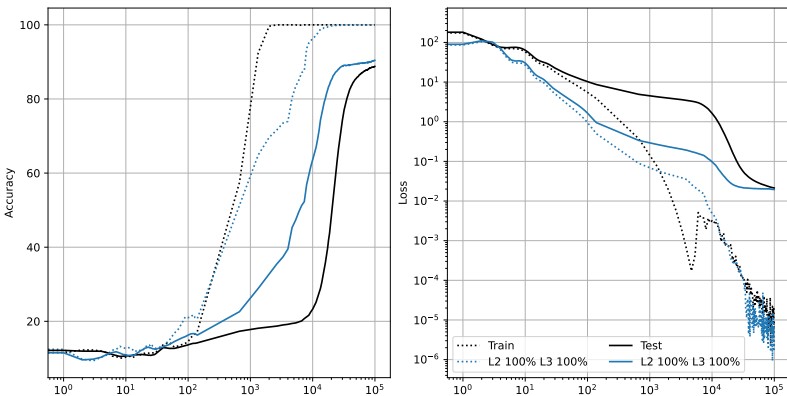

Figure 28: Train (dotted) and test (solid) accuracy with decomposed learning on Layer 2 and 3 100% of the ranks in comparison with the baseline normally trained model (black). Mean from 3 runs.

This section highlights and shows that decomposed learning can be used to avoid the grokking phenomena and improve performance on the grokking-induced MNIST task.

## G    DECOMPOSED LEARNING ON REAL-WORLD TASKS

This section explores how decomposed learning affects learning on real-world tasks. It explores a Transformer on the Tiny Shakepreare dataset and a ViT (Dosovitskiy et al., 2021) on the CIFAR10 dataset. The general finding is that decomposed learning can improve performance marginally, with the most significant increase for the Shakespeare dataset being 0.2468%, and reduce the model's size while maintaining similar performance. Using 50% of the ranks for all layers resulted in a compression ratio of 0.7215 and a reduced performance difference of 0.1448% while having a smaller generalisation gap than the baseline model. For CIFAR 10, decomposed learning improved the model's performance by 2.97% if all layers were decomposed 100% and could achieve a compression ratio of 0.4394 with a performance degradation of 1.68% if all layers expect the output and feed-forward block fc1 where decomposed to 12.5% of the full rank for the respective layer. This section highlights that decomposed learning can improve model performance and reduce model size on real-world tasks with limited affect on the performance.

### G.1    TINY SHAKESPEARE

To explore how decompose learning affects models on real-world tasks, we train a 6-layer transformer decoder architecture with a width of 384 and 6 attention heads, with a dropout of 0.2, after key, value and query layers and projection layer in the attention heads and after the last layer in the feed-forward blocks. The model consists of 10.788929 million training parameters and is trained on the TinyShakespeare dataset, which consists of 40,000 lines of Shakespeare from a variety of Shakespeare's plays, with a batch size of 64, a block size of 256, a learning rate of 3e-4. It is optimised with the Adam optimiser for 5000 iterations.

Decomposed learning is performed on the position embedding, token embedding, multi-head-attention head (key, value, and query layers), multi-head projection, feed-forward blocks, and the output layer independently with all other layers trained normally and all layers together, with 12.5%, 25%, 50%, 75% and 100% of the full rank for the respective layer, with the best test accuracy recorded. The model's performance with decomposed learning is explored using the first 10%, 50% and 90% of the dataset, with the last 90%, 50% and 10% used for testing the model, see Figure 29. The model achieves a baseline accuracy with 10% 50%, and 90% is 43.6540%, 54.4636% and 58.1418%, respectively, showing that increasing the training data improves the model's performance as expected.

The same general trend, that as the dataset increases, fewer ranks can be used to maintain performance follows albeit only when using 50% and 90% of the training data, Figure 29 and 90%, Figure 29c, as even though the decrease in performance is more with all layers at 12.5% with 90% training data than with 50% of the training data, the test accuracy for 90% training data is higher than 50% training data, reporting 51.8475% and 50.9275% respectively. It is also important to note that the generalisation gap, the difference between the train and test accuracy, Figure 30, is smaller when using 90% of the training dataset with a value of 3.3944%, whereas, with 50% of the training data, it is 5.6651%, suggesting that with more data fewer ranks can be used to reduced overfitting and maintain performance.

When observing with 10% training data, the effect is less clear. It may be an artefact of the overall worse model performance and a larger generalisation gap, especially given that decomposed learning does not improve the model's performance to match models trained with more data. For the position embedding and the multi-head attention, training with less data results in fewer ranks that can be used to maintain performance. However, for all other layers, fewer ranks result in improvements, albeit lower than if trained with more data. This result suggests that decomposed learning performs as implicit regularisation during training.

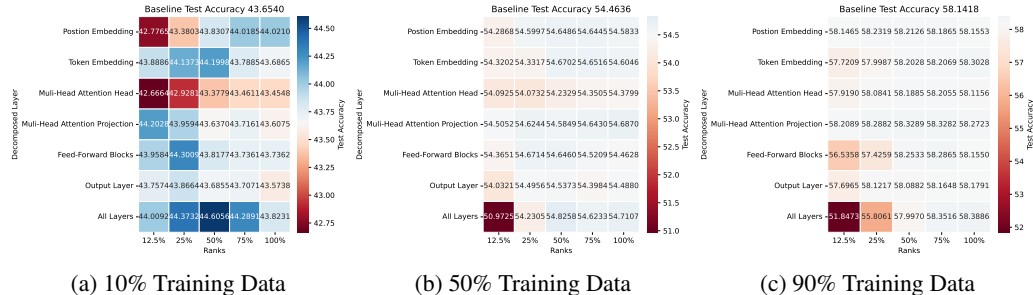

(a) 10% Training Data      (b) 50% Training Data      (c) 90% Training Data

Figure 29: Test Accuracy after Decomposed learning with 12.5%, 25%, 50%, 75% and 100% of the full rank for the respective layer. Blue cells indicate improved performance, whiteish cells indicate little change, and red cells indicate reduced performance. The mean from 3 runs is reported.

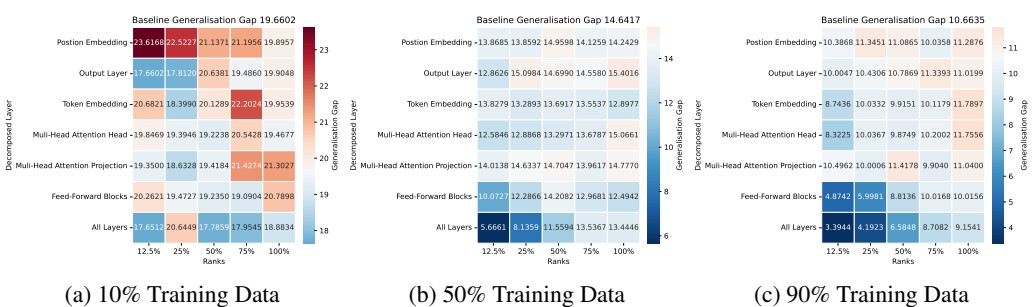

(a) 10% Training Data      (b) 50% Training Data      (c) 90% Training Data

Figure 30: Generalisation gap after Decomposed learning with 12.5%, 25%, 50%, 75% and 100% of the full rank for the respective layer. Blue cells indicate improved performance, whiteish cells indicate little change, and red cells indicate reduced performance. The mean from 3 runs is reported.

With decomposed learning, the model, when trained on 90% of the dataset, can be compressed to 0.7215 of its original size (all layers using 50% of full ranks for the respective layer) during training and achieve an accuracy of 57.9970, resulting in a performance degradation of 0.1448%. When decomposing on all layers using 12.5% of the full ranks for the respective layer, the model is compressed to 0.1683 of its original size, resulting in 1.815249 million parameters, and can achieve an accuracy of 51.8473% with a 6.2945% decrease from the baseline.

## G.2 CIFAR 10

CIFAR 10 is a 10-class image ($32 \times 32 \times 3$) classification dataset with 60000 training and 10000 test examples [CITE]. To explore decomposed learning, we train a 6-layer small-ViT with a width of 128, 4 attention heads, a forward multiplier of 2, a patch size of 4 and a dropout at 0.1. The model consists of 827,530 training parameters. It is trained on CIFAR10 with a batch size of 128, using the Adam optimiser with a learning rate of 5e-4. A linear warm-up to the learning rate of 5e-4 is used for ten epochs, followed by a cosine decay to a learning rate of 1e-5 for the remaining 90 epochs. It is trained with the following data augmentations: random crop to 32 widths and height with padding 4, random horizontal flip, Rand augmentation (Cubuk et al., 2020) and then normalised with a mean of 0.4914, 0.4822, 0.4465 and a standard deviation of 0.2470, 0.2435, 0.2616 for each respective colour channel.

Decomposed learning is performed on the position embedding, multi-head- attention head (key, value, and query layers), multi-head projection, feed-forward blocks, pre-output layer, and output layer independently with all other layers trained normally and all layers together, with 12.5%, 25%, 50%, 75% and 100% of the full rank for the respective layer, with the best test accuracy recorded, Figure 31.

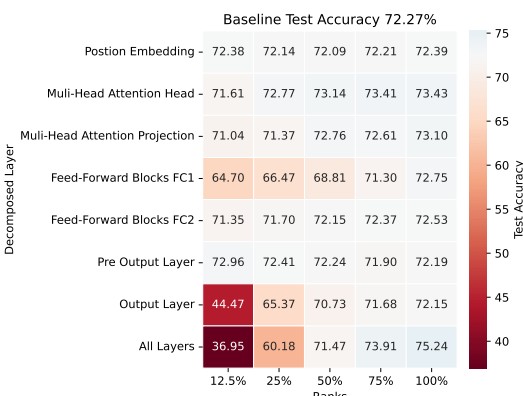

Figure 31: Test Accuracy after Decomposed learning with 12.5%, 25%, 50%, 75% and 100% of the full rank for the respective layer. Blue cells indicate improved performance, whiteish cells indicate little change, and red cells indicate reduced performance. The mean from 3 runs is reported.

Figure 31 shows that decomposed learning can improve the performance of the model when using high ranks; it is also clear that reducing the rank can degrade performance; this is most evident in the feed-forward blocks fc1 and the output layer. When decomposed learning on the multi-head attention head with 100% of ranks, the model increases performance by 1.16%. When all layers are fully decomposed (100% of ranks), the performance is 75.23%, an increase of 2.97% above the baseline. However, when all layers are decomposed to 12.5% of the full ranks for the respective layers, the models' performance is significantly reduced to 39.95% accuracy. Interestingly, the feed-forward block fc1 and the out layer are the most affected when decomposed to a low rank.

Therefore, we explore how the model performs if these layers are kept normal and all other layers are decomposed, see Figure 32. Figure 32 shows that the out feed-forward block fc1 and output layer affected the model's performance. Ignoring both the fc1 and output layer and decomposing the rest of the layers allowed the model to be compressed to 0.4394 of the original model size, resulting in 363,658 training parameters with a test accuracy of 70.29% and performance degradation of 1.68%.

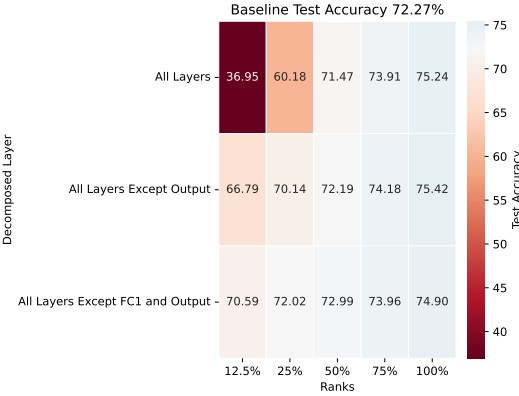

Figure 32: Test accuracy after decomposed learning with 12.5%, 25%, 50%, 75% and 100% of the full rank for the respective layer. Blue cells indicate improved performance, whiteish cells indicate little change, and red cells indicate reduced performance. The mean from 3 runs is reported.

