# Decomposed Learning and Exploring the Relationship Between Rank and Data in Grokking
# Supplementary Material

## 1 Grokking X/Y Mod 59

This section explores the grokking task of X/Y Mod 59 as this generates 3422 data samples. The same setup is used in the main body of the paper, section 4; however, it is explored using 65% and 80% of the dataset for training with, $10^6$ and $3 \times 10^5$ optimisation steps. 50% of the training dataset is not explored as little to no generalisation occurred after $10^6$ optimisation steps.

Normal training is compared against decomposed learning on only the token embedding, Figure 1, position embedding, Figure 2, multi-head attention, Figure 3, feed-forward blocks, Figure 4, output layer, Figure 5 and when decomposed learning on the token embedding, multi-head attention, feed-forward block and output layer altogether, Figure 6. The results follow the same trend as in the main body of the paper, that as more data is provided, fewer ranks can be used to mitigate and avoid grokking.

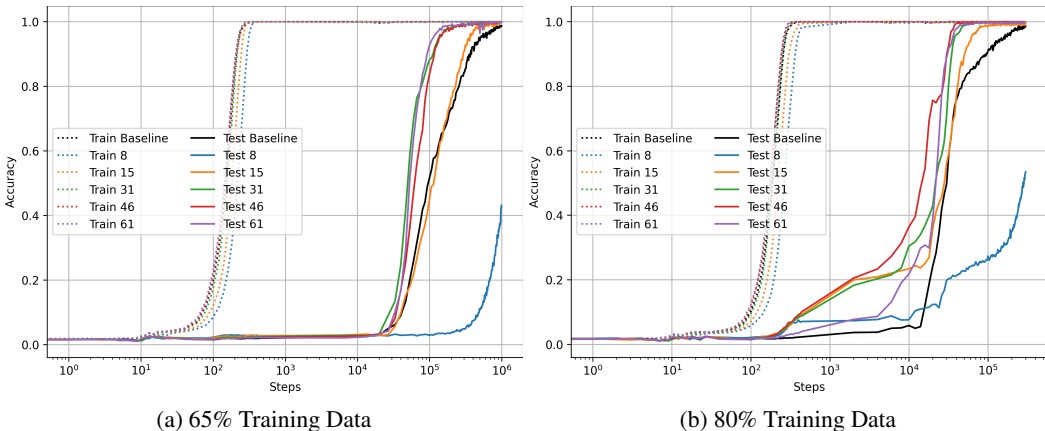

(a) 65% Training Data    (b) 80% Training Data