# OpenReview forum: "Decomposed Learning and Grokking"
_ICLR.cc/2025/Conference — Submitted to ICLR 2025_

### Official Review · Reviewer_oECT · 2024-11-03

**Soundness:** 3
**Presentation:** 3
**Contribution:** 2
**Rating:** 6
**Confidence:** 3

**Summary:**

The authors propose a method to alleviate grokking in modular arithmetic tasks by applying SVD to decompose the model’s weights. They investigate the effect of this decomposition across different network components, such as token embeddings and multi-head attention, to identify where it has the most impact. Additionally, they explore how dataset size affects grokking, finding that larger datasets and decomposing certain layers can significantly reduce the delay in generalization.

**Strengths:**

The paper is well-structured and generally clear to follow. It studies an interesting topic (at least intellectually -- but maybe not practically).

**Weaknesses:**

The paper’s motivation is not strongly established. The framing suggests that grokking is a problem to be mitigated. For example, the authors state, “...poses challenges for efficient learning…,” “...inefficiencies in how neural networks learn…,” and suggest grokking might apply to datasets like MNIST. They also highlight achieving "superior results with fewer parameters." However, grokking research is generally focused on inducing this phenomenon in artificial setups to study generalization in overparameterized models. I am not convinced that grokking is an issue requiring solution; rather, it is a phenomenon that reveals dynamics in certain models under specific conditions.

Another limitation of this paper is the inconsistency in empirical results across different network components. While the authors suggest that larger datasets enable lower ranks in decomposed learning, behaviors vary notably across network layers without sufficient explanation. These variabilities make it challenging to draw robust conclusions.

The discussion section mainly reiterates empirical results without offering deeper insights. Strengthening the paper would require further analysis and clearer connections to existing theoretical work. For instance, Kumar et al. (2024, https://arxiv.org/pdf/2310.06110) frame grokking as a transition from "lazy" to "rich" learning dynamics. Similarly, the “Dichotomy of early and late phase implicit biases” paper suggests grokking is tied to gradient flow. How would authors pose their results in existing literature? For example, the authors might consider analyzing the rate of weight or representation changes under low-rank settings or analyzing gradients and arguing that they support or disprove some of the existing perspectives. This way, they could go beyond presenting empirical results and engage in a deeper discussion.

**Questions:**

In Section 3, it would improve clarity to define the SVD dimension “r” as the true rank of matrix $A$ $(r \leq \min(m, n))$ – current notation $(r < m < n)$. Then, for $k < m$ or $k << m$, simply using $k < r$ may enhance clarity.

The Author Contributions and Acknowledgments sections appear to be copied from the ICLR template. Please remove or update them.

Minor typos: The word “grokk” should be “grok” (e.g., lines 289, 295 — "I grok" vs. "I am grok-king"), and “artefact” should be “artifact” (line 499).

---

> ### Author Response · Authors · 2024-11-26
> **Response to Review**
>
> # Reviewer oECT
>
> We thank the reviewer for the helpful and valuable review that has enabled us to interleave our results with the current literature and thus enable a better understanding of why Decomposed Learning works.
>
>
> ## Weakness 1
>
> This paper explores explicitly how learning in the decomposed representation of $U$, $\Sigma$ and $V^T$ with different layers, ranks, and amounts of training data affect the learning
> process, specifically delayed generalisation.
>
>
> It may not need a solution, but given that grokking can occur, as it is not a desirable quality in learning, it is important to see if and how it can be mitigated. The finding "superior results with fewer parameters" highlights that the method can reduce the number of parameters in the decomposed form as it still mitigates grokking, which is an interesting finding, as a smaller model size is often associated with reduced performance, not improved generalaision capabilities, as suggest by double decent and scaling scalling laws [1].
>
> [1] Kaplan, J., McCandlish, S., Henighan, T., Brown, T.B., Chess, B., Child, R., Gray, S., Radford, A., Wu, J. and Amodei, D., 2020. Scaling laws for neural language models. arXiv preprint arXiv:2001.08361.
>
>
>
> ## Weakness 2
>
> Another limitation of this paper is the inconsistency in empirical results across different network components. While the authors suggest that larger datasets enable lower ranks in decomposed learning, behaviours vary notably across network layers without sufficient explanation. These variabilities make it challenging to draw robust conclusions.
>
> We argue that this is not a weakness but a finding of the paper. We would not have expected the effect of decomposed learning to be exactly the same across layers, as the layers perform different functions within the network and have different initial ranks. However, the same general trend is found across layers: more data means fewer ranks can be used, albeit the number of ranks may differ for different layers.
>
> ## Weakness 3
>
> We conducted spectral analysis through training with the stable rank, Appendix D, which highlighted that decomposed learning can speed up the process of transitioning from a sufficiently high, stable rank to a low, stable rank if a high enough initial rank is used, which in turn allows for faster generalisation. The transition from high to low, stable rank is slow when using a normally trained model in this grokking task and may explain the delayed generalisation. This result suggests that decomposed learning helps the implicit regularisation process in reducing the stable rank more effectively and thus can reduce the steps required for grokking. We have then tied this with the current literature that suggests grokking a transition from "lazy" to "rich" learning dynamics Kumar et al. (2024), suggesting that lazy learning happens in higher dimensional space and feature learning happens in lower dimensional space.
>
> ## Questions
>
> Thank you for these suggestions we have incorporated into the paper.

---

> ### Author Response · Authors · 2024-12-02
>
> Dear oECT,
>
> We hope you are well.
>
> We are messaging to ask if there are any additional questions concerning our responses to your review. If there are, please let us know so we can address them.
>
> We value your feedback and the time and effort spent reviewing this work.

---

### Official Review · Reviewer_ehxC · 2024-11-04

**Soundness:** 3
**Presentation:** 3
**Contribution:** 3
**Rating:** 8
**Confidence:** 4

**Summary:**

This paper illustrates how parameterizing the layer in neural networks, using SVD decomposition can mitigate the phenomemon of grokking to some extent. Grokking refers to the phenomenon where neural networks achieve perfect training accuracy, far before they achieve greater than random test accuracy. This paper conducts a detailed empirical study using a simple 2-layer transformer and a simple task i.e. modular arithmetic and investigate the effects of decomposing the weight matrix using SVD decomposition at the initialization. The approach is to decompose the weight matrix after initialization into U S V matrices, reduce rank by having a sparse S matrix and not explicitly preserve the SVD decomposition during training. They conduct comprehensive experiments on the different components of the 2-layer transfomer (token embedding / positional embedding / feed-forward / multi-head attention / output layers) independently and together, varying the rank & volume of training data used.

**Strengths:**

- Systematic and scientific approach to studying the problem of grokking: the paper does a careful controlled study of the effect of rank on various components.
- Illustrating conclusively that there are strong correlations between decomposing layers into U S V  and mitigating the phenomonen of grokking.

**Weaknesses:**

- Insufficient discussion of connections to prior work: the idea of leveraging SVD decomposition for better generalization as well as more parameter-efficient learning has been discussed in several different bodies of work in ML: pruning (Compressing Neural Networks: Towards Determining the Optimal Layer-wise Decomposition etc.), low-rank gradients (GaLore: Memory-Efficient LLM Training by Gradient Low-Rank Projection etc.: to name a few.  A thorough discussion of this related work will help place this paper appropriately in the current body of work.

**Questions:**

Since the decomposed learning doesn't explicitly preserve the orthogonality of the columns of U and V, I would be curious to know what the structure of the low-rank decomposed is at the end of the training. In particular,
1. Do they retain some orthogonality between columns throughout training, are there any trends here?
2. The paper mentions the need for "high rank" decompositions. Can the authors verify, that training with this high rank, truly "uses" the full rank i.e. the final weight matrix has rank = the constraint placed by the decomposition? If not, this might point to further inefficiencies in training (such as the one the authors discover) that may contribute towards grokking?

---

> ### Author Response · Authors · 2024-11-26
> **Response to Review**
>
> # Reviewer ehxC
>
> We thank reviewer ehxC for their concise review and highlighting the missing papers concerning the related work, which has been added to further support the paper's investigation. We also thank them for their questions.
>
> # Weakness 1
>
> We have updated the releated to highlight the connection to using SVD to improve generalisation within the current literature.
>
> # Question 1 and 2
>
> In Appendix C, we show the orthogonality of U and V after training on the token embedding, which shows that the layers do not retain orthogonality between columns. We also show that the model retains the final rank constraint set in the original training.

---

> > ### Comment · Reviewer_ehxC · 2024-11-26
> >
> > Thank you for your response to my review. I am happy to stick with my recommendation for acceptance.
> >
> > With regards to Q2, my question was whether you can confirm that the weight matrices actual rank is indeed equal or close to the rank constraint set. It is possible that the weights do not even utilize the rank specified by the constraint.

---

> ### Author Response · Authors · 2024-11-29
>
> Thank you. We very much appreciate this.
>
> As to Q2. We am sorry but we am not sure what you are asking with this:
> `weight matrices actual rank is indeed equal or close to the rank constraint set.`  do you mean:
>
> 1. The weight matrices of the baseline model equal or close to the rank constrained of the decomposed model. i.e is the baseline models weights approximately rank 12?
> 2.  To check if the rank constrained, 12 for example, results in a with weight matrix that is rank 12 or lower?
>
> We hope the following provides some clarity to perspectives 1 and 2 of the question:
>
>
> The singular values are an effective method to calculate the rank of a matrix; when the singular value is zero, this indicates a rank deficiency at that row/column.
>
> For example, if we have a matrix
> \\[
> A = \\begin{bmatrix}
> 1 & 2 & 3 \\\\
> 7& 9 & 22 \\\\
> 2 & 4 & 6 \\\\
> \\end{bmatrix}
> \\]
>
> And perform $SVD$ on $A$, we get the following singular values to 5. d.p.
>
> $$\\Sigma = [26.109, 1.52, 0] $$
>
> Which shows it is rank 2. This finding is also evident in matrix $A$; row 3 is two times row 1, which makes it rank 2.
>
> The token embedding in Appendix C shows that all decomposed forms use the total ranks available to them, although rank 99 (brown) does start to have lower singular values than the baseline (blue) past index ~77.

---

### Official Review · Reviewer_NchR · 2024-11-04

**Soundness:** 2
**Presentation:** 3
**Contribution:** 2
**Rating:** 3
**Confidence:** 4

**Summary:**

This paper introduces a method called Decomposed Learning, which modifies the weight matrices of neural networks using Singular Value Decomposition (SVD), in an effort to investigate the grokking phenomenon and its connection to the structure of the weights. There is a growing body of work suggesting that grokking is linked to poor training setups; conversely, previous research has also explored different weight-matrix decompositions and their impact on training dynamics and efficiency, albeit in larger-scale, non-grokking contexts.

The authors apply their Decomposed Learning method to Transformers trained on the task of division mod 97, which is known to exhibit grokking under certain hyperparameter settings. Through empirical evaluations, the authors demonstrate that Decomposed Learning can be applied to different weight matrices of a Transformer, reducing or even eliminating grokking. The authors further study the effect of rank reduction in the decomposition, finding that different ranks of SVD can significantly affect the efficiency and generalization capability of learning, especially when coupled with different training set fractions.

**Strengths:**

1. The experimental section is clear and straightforward, and the results are easy to interpret. The experimental setup is well-designed and thorough within its *specific context*, that is, the single task of division mod 97. The authors evaluate Decomposed Learning on that single task and systematically apply different ranks to various Transformer layers. This consistent and controlled analysis provides good empirical evidence to support the claims about mitigating grokking on this task.
2. The authors find a simple strategy (SVD-based decomposition of the weight matrices) to mitigate grokking on their specific task.
3. If Decomposed Learning can be extended beyond this simple setting, it could be impactful for improving the training and efficiency of
models.

**Weaknesses:**

While this paper has an interesting result in that it finds a simple SVD-based strategy to mitigate grokking in this toy setting, I think that some more work is needed to justify the broader claims.

1. The paper does not thoroughly position itself in the broader context, so it is somewhat difficult to assess the contribution in relation to prior works. Explicitly discussing how Decomposed Learning differs from or advances previous techniques would be helpful. **Comparing and contrasting**, providing more explicit comparisons to prior methods using SVD or other decomposition techniques. This is important since SVD and other decomposition methods have been previously used for dimensionality reduction, parameter efficiency and reducing training times.
2. The paper has limited experimental scope (single task of division mod 97) despite claiming that the method has implications for Transformers more broadly. The method should be tested on more realistic tasks to see whether it offers the same benefits, otherwise it is hard to generalize the findings to broader tasks (e.g. vision, NLP).
3. The introduction of SVD and its impact on grokking isn't explained in a very thorough way. More intuition on why this decomposition works (beyond the empirical results) and why it was chosen as opposed to other decompositions could be beneficial. More theoretical justification could be useful.
4. The relevance of grokking in practical settings is unclear. Section 4 seems to imply that it can be induced for MNIST, but it is "contrived" and "forced". Is grokking a widespread phenomenon in practical settings then or not? Are there works showing how widespread and relevant it is? It would be important to answer this to better understand the broad applicability of your method, since it focuses on grokking settings exclusively.
> Decomposed learning is explored in grokking using the division mod 97 task matching the original experimental setup by Power et al. (2022). This task is explored as it is the foundational grokking experiment and, therefore, is the most appropriate case to explore as artificial cases, such as grokking induced MNIST (Liu et al., 2023), could impact the training mechanisms as it is contrived and forced example.
5. Even if your paper will only focus on the specific phenomenon of grokking, it would be important to show how your method works on these other examples (e.g. MNIST) which *are* known to grok.

**Questions:**

1. Did you compare SVD to other decomposition methods? What made you consider SVD specifically? Could you provide more intuition on why the SVD decomposition specifically aids in reducing grokking? It would be helpful to understand the theoretical basis behind this effect.
2. Did you test your Decomposed Learning method on non-grokking tasks to see how it would affect training dynamics, sample efficiency and performance? This would also give insight on whether the method is applicable more broadly outside the context of grokking (since it's not clear how widespread and relevant grokking is in the first place). Additionally, how sensitive is your method to hyperparameter settings? Grokking itself is sensitive to hyperparameter settings.
3. Is this method applicable to larger and more varied datasets beyond the current setup?
4. Would you consider grokking to be a form of "slow" learning speed due to suboptimal training conditions? If so, would mitigating grokking  mean that your method speeds up training? How would this differ from other decomposition methods that have been shown to speed up training? E.g. the paper says:
> This suggests the strengths of changing the representation of the weight matrix to ease training, which is supported by work by Paul & Nelson (2021), who proposed a learning method using SVD on dense linear layers to reduce the rank progressively and, by extension, the dimensionality of the network during training. This method reduced the training times up to 50% with minimal impact of accuracy on audio classification problems.

---

> ### Author Response · Authors · 2024-11-26
> **Response to Weakness**
>
> We thank the reviewer for the time taken and the carefully outlined feedback.  We have taken it on board and added substantial information to the appendix because of it, which we believe has helped improve the paper's quality.
>
> ## Response to Weakness 1
>
> The aim of the paper was to gain an understanding of how the rank of the layers in a neural network and the amount of data affect delayed generalisation by decomposing layers into U S V and fixing the rank instead of exploring if this is the best method to reduce or remove the grokking phenomenon. We do believe investigating other decomposition methods would be an interesting line of enquiry, but we do not have sufficient time to do it within this rebuttal period.
>
>  ## Response to Weakness 2
>
> Thank you for this recommendation. In Appendix G, we apply decomposed learning to a transformer on the Shakespeare dataset, with the model able to achieve an improvement in performance of 0.2468% and being able to compress the model with a compression ratio of 0.7215 and a reduced performance difference of 0.1448% while having a smaller generalisation gap than the baseline model. We also trained a ViT on CIFAR10 and improved performance by 2.97%  and could achieve a compression ratio of 0.4394 with a performance degradation of 1.68%. This section highlights that the general findings could be extended to Transformers more broadly.
>
>
> ## Response to Weakness 3
>
> The intuition/idea behind using SVD is that learning the matrics $U$, $\\Sigma$ and $V^T$, which can be linearly combinded to create $A$, is easier than learning $A$. Because $U$, $\\Sigma$, and $V^T$ represent sub-problems to optimise and, thus, hopefully, easier to learn. This is synonymous with the divide-and-concur algorithm of breaking problems down into simple sub-problems that are easier to solve. SVD is used as it is straightforward to implement and truncate and can be applied to non-square matrices, which are common in neural networks. Exploring how other decomposition methods affect grokking would be an interesting line of inquiry. However due to time and computational reasons, we could not explore this in the rebuttal period.
>
> We conducted spectral analysis through training with the stable rank, Appendix D, which highlighted that decomposed learning can speed up the process of transitioning from a sufficiently high, stable rank to a low, stable rank if a high enough initial rank is used, which in turn allows for faster generalisation. The transition from high to low stable rank is slow when using a normally trained model in this grokking task and may explain the delayed generalisation. This result suggests that decomposed learning helps the implicit regularisation process in reducing the stable rank more effectively and thus can reduce the steps required for grokking. Please read Appendix D for a more thorough explanation and explanation as to why decomposed learning is able to mitage grokking.
>
>
> ## Response to Weakness 4
>
> We are unaware of any works showing how widespread grokking is in real-world tasks. We expand the findings to real-world tasks in Appendix G, applying decomposed learning to a transformer on the Shakespeare dataset and a ViT on the CIFAR 10 dataset, although we are not trying to achieve state-of-the-art results here but simple show that the method works.
>
> ## Response to Weakness 5
>
> We have shown that the method is able to mitigate the grokking phenomena on MNIST in Appendix F.

---

> > ### Comment · Reviewer_NchR · 2024-12-02
> > **Official Comment by Reviewer NchR**
> >
> > I thank the authors for their responses, for providing intuition on why this method was chosen and for conducting additional experiments. I appreciate the extra experiments, which adds to the soundness of the work, but the contribution is still unclear to me.
> >
> > **Response to Weakness 1**
> > > The aim of the paper was to gain an understanding of how the rank of the layers in a neural network and the amount of data affect delayed generalisation by decomposing layers into U S V and fixing the rank instead of exploring if this is the best method to reduce or remove the grokking phenomenon. We do believe investigating other decomposition methods would be an interesting line of enquiry, but we do not have sufficient time to do it within this rebuttal period.
> >
> > I appreciate the revision to the related work section 2.1 by pointing out other works that used SVD. It would be helpful to write more about how these differ from your work, by comparing and contrasting them. Regarding the importance of grokking, it is still difficult to assess the contributions of the work. At least a more thorough investigation and review of the literature would be useful. E.g. how widespread is grokking?
> >
> > **Response to Weakness 4**
> > > We are unaware of any works showing how widespread grokking is in real-world tasks. We expand the findings to real-world tasks in Appendix G, applying decomposed learning to a transformer on the Shakespeare dataset and a ViT on the CIFAR 10 dataset, although we are not trying to achieve state-of-the-art results here but simple show that the method works.
> >
> > I think it would be beneficial for the authors to provide more evidence of how widespread grokking to better motivate their work, since the main claim is about mitigate grokking. This would help better position the work. I appreciate the experiment on MNIST since it's another example of grokking. I also appreciate the additional experiments on Transformers, but do they exhibit grokking on these tasks? If the method is 1) a practical mitigation strategy, then it's important to understand how widespread the phenomenon  is. If instead this work 2) aims to understand grokking from a scientific perspective, then more analysis would be required from this point of view. Depending on which point of view the authors are taking, then it requires a different approach, but the authors should be more clear which perspective they're taking and provide more thorough analysis in either case.

---

> > > ### Author Response · Authors · 2024-12-03
> > >
> > > Thank you for taking the time to take part in the discussion period.
> > >
> > > We hope the following answers the questions raised; please follow up if further clarity is required.
> > >
> > > Our aim is to understand how the amount of data and rank play a role in delayed generalisation. In retrospect, we should have titled the paper something like `Decomposed Learning and Exploring the Relationship Between Rank and Data in Grokking` to clarify this. As stated in the revised paper, more reasoning as to why this mod 97 was explored explicitly is that it is `a complete algorithmic dataset that fully represents the problem space, meaning that training on x% of the dataset represents x% of the problem space. This property allows for a precise investigation of how the amount of training data and rank affects the learning process as it is a complete problem` that can achieve perfect or near-perfect accuracy; as far as we are aware, no actual real-world tasks exhibit this property, without this property exploring this relationship between rank and data is unclear and not straightforward as the datasets are not complete representations of the solution space.
> > >
> > > The research questions are:
> > >
> > >  1. How does the decomposed representation of the weight matrix, A, affect training?
> > >  2. What is the relationship between the rank of a weight matrix and the amount of training
> > > data?
> > > 3. How are different layers affected by the decomposition and rank?
> > >
> > > The primary and core contributions/findings are as follows:
> > >
> > > 1. Different layers can learn with varying degrees of rank reduction while preserving performance and reducing/avoiding grokking using our SVD-based decomposed learning method.
> > >
> > > 2. As more training data is represented, fewer ranks are needed to mitigate or prevent the grokking phenomenon.
> > >
> > > 3. Representing the weight matrix as the product of the three matrices $U_k$, $\Sigma_k$, and $V_k^T$ improves performance and can achieve superior results with fewer parameters in this grokking setup.
> > >
> > >
> > > We are trying to scientifically understand how the amount of data and rank affect delayed generalisation; the most straightforward way to explore this is to decompose $A$ into $U_k$, $\Sigma_k$ and $V_k^T$, where $k$ is the rank. This property allows for systematically exploring how the rank $k$ affects learning as it is fixed. We are not trying to state that this is the best or most effective way to reduce the grokking phenomenon. We are exploring ` how data and rank affect delayed generalisation`. Therefore we do not understand how comparing and contrasting these methods provides a helpful background to our exploration.
> > >
> > > However, to clarify this, we provide the following, which will be added to the paper upon acceptance in a form that makes it clearer (we cannot change the paper at the moment).
> > >
> > > The main difference between our method and the methods mentioned in the background is that we perform SVD at the initialisation and set the rank which is then maintained throughout training, this is applied to the weights and allows training $U$ and $\Sigma$ and $V^T$ as separate components without retaining the SVD properties of orthonormality and diagonality.
> > >
> > > The LoRA method (Hu et al., 2022) is used for finetuning and trains a weight matrix as a rank-reduced composition of two matrices, initialised with Gaussian distribution for the first matrix and zeros for the second matrix. The OFT  method (Qiu et al., 2023) finetunes with a ranked reduced matrix that is orthogonal to the matrix being finetuned. The LoKa method (Edalati et al., 2022) finetunes the Kronecker product of two matrices. LoHa (Hyeon-Woo et al., 2023) uses the low-rank Hadamard product of two matrices to reduce parameters during training and allow for more efficient updates during federated learning. The work by Zhao et al. (2024) and Zhang et al. (2024) applies SVD to the gradient updates with a fixed rank. The work by Swaminathan et al. (2020) and Liebenwein et al. (2021) is applied post-training to compress the size of the network. Paul & Nelson (2021) dynamically change the layers' rank through training, periodically recomposing the matrix, and performing SVD to reduce the rank of the matrix and continue training.
> > >
> > > Our method is the only method that performs SVD at the start and then maintains the rank selected throughout training. This method allows for the straightforward and effective analysis of how the amount of data and rank affect delayed generalisation, as the rank does not change through training.

---

> > > > ### Author Response · Authors · 2024-12-03
> > > >
> > > > We want to clarify that although decomposed learning can reduce the time it takes to generalise, it is a finding but not a research goal. We, therefore, think the question of how often this grokking occurs is not relevant to our exploration of how data and rank affect delayed generalisation, which can be considered interesting in and of itself regardless of how often grokking occurs given that grokking has been shown to occur. As stated in our response to weakness1, the central claim is not about mitigating grokking, but instead to `gain an understanding of how the rank of the layers in a neural network and the amount of data affect delayed generalisation by decomposing layers into U S V and fixing the rank instead of exploring if this is the best method to reduce or remove the grokking phenomenon` we also state in the abstract `These results suggest that our SVD-based method provides a practical and scalable solution for mitigating grokking, with implications for broader transformer-based learning tasks` with the key word being **suggest** that it `provides a practical and scalable solution for mitigating grokking` which again is not a strong claim but an observation.  **We are exploring how rank and data affect delayed generalisation**, which is neither point one (`a practical mitigation strategy,`) nor exactly point two  (`aims to understand grokking from a scientific perspective`), but its **own** important question.
> > > >
> > > > In the supplementary material, we also provided another grokking task, Mod 59, which exhibits the same findings as the main body. The experiments on CIFAR10 and Tiny Shakespear with transformers do not exhibit grokking as your request was `The method should be tested on more realistic tasks to see whether it offers the same benefits, otherwise it is hard to generalise the findings to broader tasks (e.g. vision, NLP).` therefore we did not look for large scale tasks that grok but instead real-world tasks to show that the same benefits are realised, such that the work could generalise to broader tasks as requested, which we show it does.
> > > >
> > > > ### References
> > > >
> > > > Ali Edalati, Marzieh Tahaei, Ivan Kobyzev, Vahid Partovi Nia, James J. Clark, and Mehdi Rezagholizadeh. Krona: Parameter efficient tuning with kronecker adapter, 2022. URL https:
> > > > //arxiv.org/abs/2212.10650.
> > > >
> > > > Edward J Hu, yelong shen, Phillip Wallis, Zeyuan Allen-Zhu, Yuanzhi Li, Shean Wang, Lu Wang,
> > > > and Weizhu Chen. LoRA: Low-rank adaptation of large language models. In International Conference on Learning Representations, 2022. URL https://openreview.net/forum?id=nZeVKeeFYf9.
> > > >
> > > > Nam Hyeon-Woo, Moon Ye-Bin, and Tae-Hyun Oh. Fedpara: Low-rank hadamard product for
> > > > communication-efficient federated learning, 2023. URL https://arxiv.org/abs/2108.06098.
> > > >
> > > > Lucas Liebenwein, Alaa Maalouf, Dan Feldman, and Daniela Rus. Compressing neural networks:
> > > > Towards determining the optimal layer-wise decomposition. Advances in Neural Information
> > > > Processing Systems, 34:5328–5344, 2021. URL https://proceedings.neurips.cc/paper_files/paper/2021/file/2adcfc3929e7c03fac3100d3ad51da26-Paper.pdf.
> > > >
> > > >
> > > > Vlad S Paul and Philip A Nelson. Matrix analysis for fast learning of neural networks with application to the classification of acoustic spectra. The Journal of the Acoustical Society of America, 149(6):4119–4133, 2021. URL https://pubs.aip.org/asa/jasa/article/149/6/4119/1059327/Matrix-analysis-for-fast-learning-of-neural.
> > > >
> > > > Sridhar Swaminathan, Deepak Garg, Rajkumar Kannan, and Frederic Andres. Sparse low rank
> > > > factorization for deep neural network compression. Neurocomputing, 398:185–196, 2020. URL https://www.sciencedirect.com/science/article/pii/S0925231220302253.
> > > >
> > > >
> > > > Zeju Qiu, Weiyang Liu, Haiwen Feng, Yuxuan Xue, Yao Feng, Zhen Liu, Dan Zhang, Adrian
> > > > Weller, and Bernhard Scholkopf. Controlling text-to-image diffusion by orthogonal finetun- ¨
> > > > ing. In Thirty-seventh Conference on Neural Information Processing Systems, 2023. URL
> > > > https://openreview.net/forum?id=K30wTdIIYc.
> > > >
> > > > Zhenyu Zhang, Ajay Jaiswal, Lu Yin, Shiwei Liu, Jiawei Zhao, Yuandong Tian, and Zhangyang
> > > > Wang. Q-galore: Quantized galore with int4 projection and layer-adaptive low-rank gradients,
> > > > 2024. URL https://arxiv.org/abs/2407.08296.
> > > >
> > > > Jiawei Zhao, Zhenyu Zhang, Beidi Chen, Zhangyang Wang, Anima Anandkumar, and Yuandong
> > > > Tian. Galore: Memory-efficient llm training by gradient low-rank projection, 2024. URL https://arxiv.org/abs/2403.03507.

---

> ### Author Response · Authors · 2024-11-26
> **Response to Questions**
>
> We thank the reviewer for the time taken and the carefully outlined feedback. We have taken it on board and added substantial information to the appendix because of it, which we believe has helped improve the paper's quality.
>
> ## Question 1
>
> The intuition behind using SVD is that learning the matrics $U$, $\\Sigma$ and $V^T$, which can be linearly compared to creating $A$, is easier than learning $A$. Because $U$, $\\Sigma$, and $V^T$ represent sub-problems to optimise and, thus, hopefully, easier to learn. This idea is synonymous with the divide-and-concur algorithm of breaking problems down into simple sub-problems that are easier to solve. SVD is used as it is straightforward to implement and truncate and can be applied to non-square matrices, which are common in neural networks. Exploring how other decomposition methods affect grokking would be an interesting line of inquiry. However due to time and computational reasons, we could not explore this.
>
> We conducted spectral analysis through training with the stable rank, Appendix D, which highlighted that decomposed learning can speed up the process of transitioning from a sufficiently high, stable rank to a low, stable rank if a high enough initial rank is used, which in turn allows for faster generalisation. The transition from high to low stable rank is slow when using a normally trained model in this grokking task and may explain the delayed generalisation. This result suggests that decomposed learning helps the implicit regularisation process in reducing the stable rank more effectively and thus can reduce the steps required for grokking. Please read Appendix D for a more thorough explanation and explanation as to why decomposed learning is able to mitage grokking.
>
>
> ## Questions 2 and 3
>
> In Appendix G, we apply decomposed learning to a transformer on the Shakespeare dataset, with the model able to achieve an improvement in performance of 0.2468% and being able to compress the model with a compression ratio of 0.7215 and a reduced performance difference of 0.1448% while having a smaller generalisation gap than the baseline model. We also trained a ViT on CIFAR10 and improved performance by 2.97%  and could achieve a compression ratio of 0.4394 with a performance degradation of 1.68%. This highlights that the general findings could be extended to Transformers more broadly.
>
> We also explored how Decomposed Learning and weight decay interacted and found that using weight decay on the decomposed layers resulted in worse performance see Appendix E.
>
> ## Questions 4
>
> Yes, we would consider grokking to be a slow form of learning; this is supported by Appendix D, which shows that in the grokking condition, the model takes longer to reduce the stable rank across layers, but in decomposed learning, this happens quicker. For the case of grokking, it can be viewed as speeding up training; Appendix A.2 shows that decomposed learning can result in 61.67 times fewer steps to reach a 1% generalisation gap than conventional training. We do not compare directly to Paul & Nelson (2021) as the paper is not trying to be competitive with the state of the methods but instead gain a better understanding of how the amount of training data and model rank play a role in delayed generalisation.

---

> ### Author Response · Authors · 2024-12-02
>
> Dear NchR,
>
> We hope you are well.
>
> We are messaging to ask if there are any additional questions concerning our responses to your review. If there are, please let us know so we can address them.
>
> We value your feedback and the time and effort spent reviewing this work.

---

### Official Review · Reviewer_hNTw · 2024-11-04

**Soundness:** 2
**Presentation:** 3
**Contribution:** 2
**Rating:** 3
**Confidence:** 4

**Summary:**

This paper studies the phenomenon of grokking in the context of “decomposed learning” which seeks to optimize the layers of a neural network as independent matrices given by the singular value decomposition of each layer. The authors study decomposed learning in two layer transformers while varying the rank of different layers in the model and present experiments to show when this method works and when it does not.

**Strengths:**

The authors find that in some cases the total number of parameters can be reduced, as can the rank of each layer, while achieving similar or faster generalization speed in grokking modular division with mod 97, indicating that fewer parameters can be optimized in total while still leading to a generalizable model, which is a desirable thing.

**Weaknesses:**

Overall I am a bit confused as to the takeaway suggested here. In most cases it seems that increasing the amount of training data enables lower rank decompositions to be as good or maybe a little better than the baseline. On the other hand when 50% of the training data is used it seems that, for the most part, higher-rank decompositions (many of which increase the parameter count except in a couple cases i.e. in the token embeddings for rank 25) generalize faster and lower rank decompositions generalize slower with respect to number of optimization steps. However when increasing the amount of training data it is well known that the delayed grokking phenomenon becomes less delayed and in some cases even vanishes, so I am not convinced it is fair to say that decomposed learning becomes effective at removing grokking at low ranks in these settings if the delayed generalization phenomenon is not even clear there. In those cases maybe we can conclude that low ranks are just as viable as higher ranks when there is sufficient data, and that could help lower the total parameter count needed to train, but I’m not sure what this says about grokking as a phenomenon? Rather a note on some relationship between the amount of training data and parameter count.

I think there are potentially interesting experiments in this paper that could be suggestive of nice principles in deep learning, feature learning, and specifically grokking, but the way it is presented and the conclusions drawn seem quite unclear to me and I think this manuscript would benefit from a rewrite or more clarity as to what is the core argument being made. In its current form I’m not sure if I’ve learned much about why or what is happening to cause grokking, nor have I learned much about when or why low rank adaptations work and what they are promoting (some notion of complexity is missing that is perhaps being optimized for in the decomposition? Or something else? It’s not clear to me at least.)

While the experiments work for modular division mod 97, it seems fairly reasonable and simple to change the prime number as well when varying the amount of training data to further examine data sparse settings. For instance 50% of training data at mod 31 is a lot less data than 50% of training data at mod 97. Maybe these variations don’t impact your conclusions, but I think in the case of training a two layer transformer on <= 31^2 total samples it should be a simple and fast experiment to run, even on a CPU or cheap GPU. However, I understand that experimental work is compute-limited so my main concerns are less about this lack of some experiments rather about the core narrative story being told.

My last note would be that it would be really interesting to study spectral properties of the recomposed weight matrices as well as the decomposed matrices U, Sigma, V^T after being optimized. The authors note that the SVD decomposition leads to three matrices that are independently optimized when training the network, and that they do not enforce that the columns of U or V are orthonormal nor do they enforce that Sigma is still diagonal. It would be really interesting to track some rank measure of A and the reconstituted A (i.e. stable rank can measure this), as well as plot some measures of rank of the individual matrices in a layer that are being optimized. In general there are a lot of things that should still be studied in this setting to really understand what is going on.

See Questions for further discussion.

**Questions:**

In many of the experiments in Power et al. (2022) they use weight decay = 1 but note that there are some results presented using weight decay = 0 when increasing the number of optimization steps. As far as I can tell you are using weight decay = 0 throughout, can you comment on the choice and any comparisons? In particular, how could weight decay affect the low-rank decomposition? This seems like an important part of the story given that there is a fair amount of uncertainty as to the role or necessity of weight decay in grokking (or lack thereof in some cases).

You mention a few times that the experimental evidence supports the idea that training the decomposed version of A allows for “more complex transformations to be learned more efficiently” due to there being cases when generalization happens faster than the baseline in a decomposed learning setting and yet there being fewer parameters total. I’m not totally sure what “more complex transformations” or “more efficiently” in this context means. What is the notion of complexity and efficiency that you are using in the context of grokking? Is it possible that simpler transformations are being learned faster? I think this manuscript is missing a fair but of context to make these sort of claims, and while the experiments are presented clearly it is hard to understand what exactly they mean or how it implies something about more complex transformations or more efficient learning.

As for the experiments with decomposing multiple layers simultaneously it would be great to see more ablations on how to choose how low rank you can go with different layers? For instance if I go sufficiently high rank in my token and position embeddings does it let me achieve fast generalization with even lower rank multi-head attention layers than it would otherwise? There are a lot of natural experiments that would really make this story more compelling in understanding how the rank of different layers impacts the overall learning process and interacts with the ranks of other layers.

Does Sigma ever become non-diagonal in training? If so, what does it look like? What is the rank of U, Sigma, V^T after training?

Are you training the rank-one decomposition of the A matrix and then putting it back together? Or are you training all of U, Sigma, V^T as is, in which case if Sigma becomes non-diagonal then you might end up with a reconstituted A matrix that has a higher rank than the decomposed rank suggested, if I’m not mistaken? Correct me if I’m wrong of course, but in this case it would be interesting to understand the spectral dynamics of the various matrices in play, and/or plotting something like their deviation from initialization.

I think looking into such directions will be very fruitful and lead to numerous insights that would strengthen this paper substantially.

Some line edits I caught while reading it:

Throughout: “grokk” -> “grok”

Line 230: “perfect near-perfect” → “perfect or near-perfect”

Line 238: “data, rank, 12 start to grokk” → “data, rank 12 starts to grok”

Line 289: “Training on…” → “When training on…”

---

> ### Author Response · Authors · 2024-11-26
> **Response to Weakness**
>
> We would like to thank the reviewer for the time taken and the carefully outlined feedback. We believe their insightful feedback has enabled us to improve the quality of the paper.
>
>
> ## Response to Weakness 1
> To make this case clearer, we have provided supplementary material and a study of the grokking task of division MOD 59, where the grokking phenomenon is more evident even when training on 80% of the dataset. Increasing the training data decreases the number of ranks required for the model to generalise before the baseline and reduces or mitigates grokking. Although we agree, this is more of a statement about the relationship between the amount of training data and the model's rank directly than specifically about grokking. We have updated our Discussion section, **More Data Fewer Ranks**, to reflect this.
>
> ## Response to Weakness 2
>
> We are sorry for the confusion regarding the core argument.
>
> This paper set out to explore:
>
>  - How does the decomposed representation of the weight matrix, $A$, affect training?
>
>  - What is the relationship between the rank of a weight matrix and the amount of training data?
>
> - How are different layers affected by the decomposition and rank?
>
> With the core contributions being:
>
> - Representing the weight matrix $A$ as the product of the three matrices $U_k$,  $\Sigma_k$ and $V_k^T$ improves performance and can achieve superior results with fewer parameters in this grokking setup.
>
> - As more training data is represented, fewer ranks are needed to mitigate or prevent the grokking phenomenon.
>
> - Different layers can learn with varying degrees of rank reduction while preserving performance and reducing/avoiding grokking using our SVD-based decomposed learning method.
>
> To aid and improve the understanding of how and why decomposed learning is effective at reducing delayed generalisation, we conducted spectral analysis through training with the stable rank, Appendix D, which highlighted that that decomposed learning can speed up the process of transitioning from a sufficiently high, stable rank to a low, stable rank if a high enough initial rank is used, which in turn allows for faster generalisation. This transition from high to low stable rank is slow when using a normally trained model in this grokking task and may explain the delayed generalisation. This result suggests that decomposed learning helps the implicit regularisation process in reducing the stable rank more effectively and thus can reduce the steps required for grokking. Please read Appendix D for a more thorough explanation.
>
> ## Response to Weakness 3
>
> In the supplementary, we have provided another example of grokking and decomposed learning with division MOD 59. We observe the same findings as the main body of the paper.
>
> ## Response to Weakness 4
>
> Thank you for this recommendation; Appendix D provides an investigation into the spectral properties through training and shows that decomposed learning helps in reducing the stable rank through training, giving a potential explanation of decomposed learning effectiveness, which was not previously present in the paper.
>
> In Appendix C, we also show that at the end of the training, the U and V are no longer orthogonal, and Sigma is no longer diagonal.

---

> ### Author Response · Authors · 2024-11-26
> **Response to Questions 1 - 4**
>
> We would like to thank the reviewer for the time taken and the carefully outlined feedback. We believe their insightful feedback has enabled us to improve the quality of the paper.
>
> ## Response to Question 1
>
>  A weight Decay of 0 is used in this experiment, as with a weight decay of 1, the model could not generalise in the conventional training setting. However, we explore the effect of weight decay on decomposed learning with a one hidden layer MLP with a width of 256 on MNIST in Appendix D. Appendix D highlights that decomposed learning is less effective when weight decay is used with there being a more negative effect with higher values of weight decay, and that is enhanced when all the layers are decomposed. We attribute this to the recent finding that weight decay encourages rank minimisation, and thus, using both methods induces too strong a regularisation effect on the model.
>
> ## Response to Question 2
>
> This was an oversight on our part, and this is correct that it could instead be learning simpler transformations; we have changed the manuscript to reflect this. By more efficient learning, we meant that the model is smaller and still able to achieve the same performance, and thus, there has been a more efficient use of the parameters.
>
> ## Response to Question 3
>
> Unfortunately due to time and computational constraints we are unable to provided these results in this rebuttal period. However, will add more ablation study to the Appendix upon acceptance.
>
> ## Response to Question 4
>
> Sigma does become non-diagonal through training; see Appendix C Figure 18 for a visualisation with the token embedding layer. When reconstructing U Sigma V^T, the rank of the new matrix is the rank selected at train time. For instance, the token embedding layer is decomposed to rank 12 after training. When recomposed, it will be rank 12, see Appendix C Figure 19. In addition, U and V^T become non-orthogonal as well during training, which is also shown in Appendix C.

---

> ### Author Response · Authors · 2024-11-26
> **Response to Question 5**
>
> ## Response to Question 5
>
> No, we are training a truncated U, Sigma, V^T. So  U, Sigma, and V^T are fixed at the rank of what is requested. For example, if we have a 100 by 100 matrix (A) and we decompose it using SVD and set the rank to 12, the result is U, Sigma, and V^T is 100x12, 12x12, 12x100; by doing this, the matrix rank can never go above the specified rank, it can however go below.
>
> A simple example is if we have a 3x3 matrix, A, and we perform SVD.
>
> $$
> A =
> \\begin{bmatrix}
>     0.4784 & 0.5468 & 0.2000 \\\\
>     0.3952 & 0.7155 & 0.5241 \\\\
>     0.4797 & 0.8756 & 0.6019 \\\\
> \\end{bmatrix}
> $$
>
> $$U \\Sigma V^T = SVD(A)$$
>
> $$ U =
> \\begin{bmatrix}
>  -0.4315 &   0.9002 & -0.0588 \\\\
>  -0.5769 &  -0.3254 & -0.7492 \\\\
>  -0.6936 &  -0.2894 &  0.6597 \\\\
> \\end{bmatrix}, \\Sigma =
> \\begin{bmatrix}
>  1.6780 &  0 & 0 \\\\
>  0 &  0.2320 & 0 \\\\
>  0 &  0 &  0.0142 \\\\
> \\end{bmatrix},
>  V^T =
> \\begin{bmatrix}
> -0.4572 & -0.7485 & -0.4804 \\\\
>  0.7037 &  0.0259 & -0.7100 \\\\
> -0.5439 &  0.6626 & -0.5149 \\\\
> \\end{bmatrix}
> $$
>
> If we reduce the rank to rank 2, we get
> $$ U_2 =
> \\begin{bmatrix}
>  -0.4315 &   0.9002 \\\\
>  -0.5769 &  -0.3254 \\\\
>  -0.6936 &  -0.2894 \\\\
> \\end{bmatrix}, \\Sigma_2 =
> \\begin{bmatrix}
>  1.6780 &  0 \\\\
>  0 &  0.2320 \\\\
> \\end{bmatrix},
>  V^T_2 =
> \\begin{bmatrix}
> -0.4572 & -0.7485 & -0.4804 \\\\
>  0.7037 &  0.0259 & -0.7100 \\\\
> \\end{bmatrix}
> $$
>
> Then reconstruct at rank two to get $A_2 = U_2 \\Sigma_2 V^T_2$.
>
> $$
> A\_2 = \\begin{bmatrix}
> 0.4779 & 0.5474 & 0.1996 \\\\
> 0.3894 & 0.7226 & 0.5186 \\\\
> 0.4848 & 0.8694 & 0.6067\\\\
> \\end{bmatrix}
> $$
>
> If we perform SVD on $A_2$ we get the following singular values to 5 d.p. $A_{2 \\Sigma} =  \\begin{bmatrix} 1.67800 & 0.23196 & 0 \\end{bmatrix} $
>
> Now if we add random noise to $U_2, \\Sigma_2$ and $V^T_2$, to simulate training in decomposed learning form to create
>
> $$ U^o_2 =
> \\begin{bmatrix}
>  0.2854 &  0.5555 \\\\
> -0.6419 &  0.6172 \\\\
> -0.5148 &  0.6564 \\\\
> \\end{bmatrix},
> S^o_2 =
> \\begin{bmatrix}
> 1.4056 & 0.9218\\\\
> 0.4802 & 0.2946 \\\\
> \\end{bmatrix},
>  V^{To}_2 =
> \\begin{bmatrix}
> -0.2375 & -0.6166 & -0.7502\\\\
>  1.1256 &  0.6523 & -1.5122 \\\\
> \\end{bmatrix}
> $$
>
> Then if we perform SVD on $A^o\_2 =  U^o\_2 \\Sigma^o\_2 V^{To}\_2$ we get the following singular values to 5 d.p. $A^o_{2 \\Sigma} =  [1.81370, 0.02200, 0]$
>
> This works because the decomposed matrix shapes, we do not operate on the final column of $U$ or the final row of $V$ as we truncate the matrix. Thus, the rank is implicitly reduced and cannot be increased irrespective of the values inside $U_k$ $\\Sigma_k$ and $V^T_k$.
>
> To make clear, the full matrix form of  $U^o_2 \\Sigma^o_2 V^{To} _2 $ is the following. Therefore, the full rank could never be reconstructed; thus, regardless of the inputs, it will always be ranked 2.
>
>
> $$ U^o\_2 =
> \\begin{bmatrix}
>  0.2854 &  0.5555 & 0 \\\\
> -0.6419 &  0.6172 & 0 \\\\
> -0.5148 &  0.6564 & 0 \\\\
> \\end{bmatrix}
> $$ $$
> \\Sigma^o\_2 =
> \\begin{bmatrix}
> 1.4056 & 0.9218 & 0\\\\
> 0.4802 & 0.2946 & 0\\\\
> 0 & 0 & 0 \\\\
> \\end{bmatrix}, $$ $$
>  V^{To}\_2 =
> \\begin{bmatrix}
> -0.2375 & -0.6166 & -0.7502\\\\
>  1.1256 &  0.6523 & -1.5122 \\\\
>  0 & 0 & 0 \\\\
> \\end{bmatrix}
> $$
>
> We agree that if trained in the full form, the rank would not be bounded to the rank selected and could increase, which is why we use the truncated form to ensure the rank doesn't change through training.
>
> We also provided spectral analysis through training in Appendix D
>
> We have changed the text in Section 3 (DECOMPOSED LEARNING) to make this clearer.

---

> ### Author Response · Authors · 2024-12-02
>
> Dear hNTw,
>
> We hope you are well.
>
> We are messaging to ask if there are any additional questions concerning our responses to your review. If there are, please let us know so we can address them.
>
> We value your feedback and the time and effort spent reviewing this work.

---

> > ### Comment · Reviewer_hNTw · 2024-12-02
> >
> > Hello, I thank the authors for taking the time to present many new experiments and results based on my feedback. I certainly think that the results with weight decay and stable rank, as an example, give me some better understanding of changes in learning dynamics when using low rank decompositions for learning.
> >
> > These new results make for entire appendix sections that aren't folded into the main-text narrative quite well yet. I think that this manuscript would benefit from a solid revision taking into account the new results and recommendations from the other reviewers, and bringing all of these ideas and intuitions together more clearly.
> >
> > Given that the remaining reviewers have yet to engage in discussion with the authors, I am open to increasing my rating to somewhere around a 4 or 4.5 (which is sadly not available as an option for me to choose). But in it's current form I still feel it is below clear acceptance quality and can be much stronger with a rewrite and submission to the next venue.

---

> > > ### Author Response · Authors · 2024-12-03
> > >
> > > Thank you for engaging in the discussion
> > >
> > > While we wanted to explore and include these things to provide greater context, it is well placed in the appendix and well signposted within the main body for people interested in further exploration. We do not think it directly adds to the core message that:
> > >
> > > - Representing the weight matrix A as the product of the three matrices $U_k$, $\Sigma_k$ and $V^T_k$ improves performance and can achieve superior results with fewer parameters in this grokking
> > > setup.
> > >
> > > - As more training data is represented, fewer ranks are needed to mitigate or prevent the
> > > grokking phenomenon.
> > >
> > > - Different layers can learn with varying degrees of rank reduction while preserving performance and reducing/avoiding grokking when using our SVD-based decomposed learning method
> > >
> > > But provides additional information on why this method works and thus is suitably placed in the appendix. Where the definition of an appendix is **"a separate part at the end of a book or magazine that gives extra information"** [1]
> > >
> > > We think that adding further hyperparameters exploration on top of the already selected layer rank and amount of training data to the main body would make the paper unclear and detract from the central exploration. We explore how decomposed learning `with different layers, ranks, and amounts of training data, affects the learning process, specifically delayed generalisation.` which is suitably explored in the main body of the paper. Therefore, how weight decay affects this method and how the stable rank changes do not add to the main story but provide additional insights into the method and should be included in the appendix as it is an exploration away from the paper's primary goal. We maintained the original hyperparameters as the original paper to maintain a fair comparison such that the exploration of rank and data could be explored effectively and fairly with the baseline.
> > >
> > > We have fulfilled most of the requirements and have appropriately placed the work in the appendix while providing appropriate signposting within the main body of the paper. Given the positive response of other reviewers, we feel a rewrite is **optional** as the paper's main point has been received **without other requests for a rewrite**.
> > >
> > > In addition, we have considered the `recommendations from the other reviewers` and provided most of the responses in the appendix, which makes sense as there were requests for further explorations instead of direct criticism of the main body. Doing this has not negatively affected the paper, but it has answered the questions posed by the reviewers and improved the paper.
> > >
> > > [1] Appendix (Book Part) | English meaning - Cambridge Dictionary Cambridge Dictionary. Available at: https://dictionary.cambridge.org/dictionary/english/appendix.

---

### Meta-Review · Area_Chair_9p6L · 2024-12-22

**Metareview:**

This paper examines the phenomenon of grokking through Decomposed Learning, a method that applies SVD to the weight matrices of neural networks, treating them as independent components. The authors explore the relationship between weight structure and grokking by analyzing how this decomposition impacts training dynamics and generalization.

Focusing on a two-layer Transformer trained on the division mod 97 task, a problem known to exhibit grokking under certain hyperparameter settings, the study investigates how varying the rank of decomposed matrices and the fraction of the training set influences learning behavior. The results demonstrate that adjusting the rank can significantly reduce or eliminate grokking.

The paper has several notable weaknesses. The writing quality is lacking, with multiple reviewers highlighting that the motivation and intuition behind the method are not presented clearly. Additionally, the paper fails to provide a detailed discussion and comparison with prior work. As reviewer NchR pointed out, the most critical issue is that the paper does not demonstrate how widespread the grokking phenomenon is in practice, nor does it include practical experiments to validate its relevance.

Furthermore, if the primary goal is to study how different layers, ranks, and amounts of training data affect the learning process, the current task, experimental scale, and analysis presented in the paper are far from sufficient to support robust conclusions about the existence or impact of grokking.

Given these issues, the submission would require substantial revisions to meet the standards for acceptance. In its current state, I cannot recommend accepting this paper.

**Additional Comments On Reviewer Discussion:**

During the rebuttal period, the authors responded late and did not provide convincing arguments to address the reviewers’ concerns, resulting in no significant score changes. Some reviewers noted that the additional content introduced during the rebuttal was difficult to integrate into the main paper, while others expressed dissatisfaction with the responses, maintaining their original score. Overall, the majority of reviewers leaned towards rejecting the paper.

---

### Decision · Program_Chairs · 2025-01-22

Reject